# Identification of novel cyclin-dependent kinase 4/6 inhibitors from marine natural products

**Abhijit Debnath** [1]*, **Rupa Mazumder**[1], **Anil Kumar Singh**[2]*, **Rajesh Kumar Singh**[2]

**1** Noida Institute of Engineering and Technology [Pharmacy Institute], Institutional Area, Greater Noida, Uttar Pradesh, India, **2** Department of Dravyaguna, Institute of Medical Sciences, Banaras Hindu University, Varanasi, India

* abhijitdebnath@outlook.in, abhijit.debnath@niet.co.in (AD); anilkumar.singh113@gmail.com (AKS)

**Data Availability Statement:** https://doi.org/10.6084/m9.figshare.27172980.v1.

**Funding:** This research was funded by the Department of Health Research, Govt. of India for financial support (BHU Project code: M-48/0179;

## Abstract

Cyclin-dependent kinases 4 and 6 (CDK4/6) are crucial regulators of cell cycle progression and represent important therapeutic targets in breast cancer. This study employs a comprehensive computational approach to identify novel CDK4/6 inhibitors from marine natural products. We utilized structure-based virtual screening of the CMNPD database and MNP library, followed by rigorous filtering based on drug-likeness criteria, PAINS filter, ADME properties, and toxicity profiles. From an initial hit of 9,497 compounds, 2,344 passed drug-likeness and PAINS filters. Further ADME filtering yielded 50 compounds, of which 25 exhibited non-toxic profiles. These 25 candidates underwent consensus molecular docking using seven distinct algorithms: AutoDockTools 4.2, idock, LeDock, Qvina 2, Smina, AutoDock Vina 1.2.0, PLANTS, and rDock. Based on these results, six top-scoring compounds were selected for comprehensive 500 nanosecond all-atom molecular dynamics simulations to evaluate their structural stability and interactions with CDK4/6. Our analysis revealed that compounds CMNPD11585 and CMNPD2744 demonstrated superior stability in their interactions with CDK4/6, exhibiting lower RMSD and RMSF values, more favorable binding free energies, and persistent hydrogen bonding patterns. These compounds also showed lower Solvent Accessible Surface Area values, indicating better compatibility with the CDK4/6 active site. Subsequent *in-vitro* studies using MTT assays on MCF-7 breast cancer cells confirmed the cytotoxic effects of these compounds, with CMNPD11585 showing the highest potency, followed by CMNPD2744.

## 1. Introduction

Breast cancer remains the most common cancer in women globally, with an estimated 2.3 million new cases in 2020 [1]. It originates in breast tissue's milk ducts or lobules and can invade nearby tissues and metastasize to distant body parts due to uncontrolled cell growth [2]. Recent advances in breast cancer research have led to the development of more targeted therapies and personalized medicine approaches, significantly enhancing outcomes for many patients [3,4].

DHR File No.R.12014/57/2022-HR), under human resource development program (Young Scientist Scheme). The funders had no role in study design, data collection and analysis, decision to publish, or preparation of the manuscript.

**Competing interests:** The authors have declared that there are no competing interests exist.

Cyclin-dependent kinases 4 and 6 (CDK4/6) have emerged as crucial therapeutic targets in breast cancer treatment, particularly for hormone receptor-positive (HR+) and human epidermal growth factor receptor 2-negative (HER2-) subtypes [5]. CDK4/6 are key regulators of the cell cycle, facilitating the transition from the G1 to the S phase and enhancing cell proliferation [6]. Their overactivation is a common issue in various cancers, including breast cancer, leading to uncontrolled cell division and tumor growth [7].

The development of CDK4/6 inhibitors has revolutionized the treatment of HR+/HER2- advanced breast cancer. Three CDK4/6 inhibitors—palbociclib, ribociclib, and abemaciclib—have been approved by the FDA, showing significant improvements in progression-free survival [8,9]. These inhibitors work by preventing cell cycle progression and inducing cell cycle arrest, providing a valuable therapeutic option for patients with advanced disease [10]. In addition to their primary mechanism of action, CDK4/6 inhibitors have shown promise in combating endocrine resistance in HR+ breast cancer [10]. They also enhance anti-tumor immunity by promoting T cell memory, suggesting potential use in combination therapies with immunotherapies [11]. The success of CDK4/6 inhibitors in HR+/HER2− breast cancer has prompted further research into their potential in other cancer types and combinations with targeted therapies, chemotherapy, and immunotherapy [12]. However, CDK4/6 inhibitors are not without side effects. Common adverse events include neutropenia, leukopenia, fatigue, and gastrointestinal issues [13–16]. Some patients may experience mild hair loss or skin rash [17], and there have been reports of elevated liver enzymes suggesting potential liver injury [18]. Less common but significant side effects include cardiovascular adverse effects such as QT prolongation and venous thromboembolism [19].

Given these challenges, there is ongoing research to identify novel CDK4/6 inhibitors with improved efficacy and reduced side effects. One promising avenue of exploration is marine natural products. Marine organisms, including algae, sponges, and microorganisms, have been recognized as rich sources of bioactive compounds with potential therapeutic applications [20,21]. The marine environment, with its exceptional biodiversity and chemical complexity, offers a vast reservoir of compounds with diverse structural features and biological activities [22]. Many marine-derived compounds exhibit potent anti-proliferative effects, making them promising candidates for cancer therapy, including as potential CDK4/6 inhibitors [23]. Recent studies have utilized advanced screening techniques and *in-silico* approaches to identify marine natural products with CDK4/6 inhibitory activity [24]. For instance, some marine-derived alkaloids and terpenes have shown promising results in initial screenings, demonstrating the ability to modulate CDK4/6 activity and induce cell cycle arrest in cancer cell lines [25]. The exploration of marine fungi has yielded several compounds with potential as CDK4/6 inhibitors. These organisms, adapted to unique marine environments, often produce secondary metabolites with novel chemical structures and biological activities not found in terrestrial counterparts [26]. While research in this area is still in its early stages, the potential of marine natural products as a source of novel CDK4/6 inhibitors is significant. These compounds may offer new chemical scaffolds for drug development, potentially leading to improved efficacy or reduced side effects compared to current synthetic CDK4/6 inhibitors [27].

This study aims to identify novel CDK4/6 inhibitors from marine natural products using a comprehensive computational approach, including structure-based virtual screening, drug-likeness filtering, ADME property prediction, toxicity profiling, consensus molecular docking, and molecular dynamics simulations. By leveraging the unique chemical diversity of marine natural products, we seek to discover new lead compounds with the potential to overcome the limitations of current CDK4/6 inhibitors and provide improved therapeutic options for breast cancer patients.

## 2. Materials and methods

### 2.1 Identification of hits by virtual screening

**2.1.1 Target preparation.** The 3D X-ray crystallographic structure of the CDK4/6 complex (PDB identifier: 5L2S) was retrieved from the RCSB Protein Data Bank [28]. This structure, resolved at 2.27 Å, provides high-resolution atomic coordinates of the protein complex [29]. Before docking, the structure underwent preprocessing to optimize it for molecular modeling studies. Utilizing the Dockprep module integrated within the UCSF Chimera software package [30], the structure was systematically curated. This curation process involved the removal of covalently bound ligands, solvent compounds, cofactors, and ionic species, which could potentially interfere with subsequent *in-silico* analyses. Additionally, polar hydrogen atoms were added to the structure, a critical step for accurate representation of potential hydrogen bonding interactions in molecular docking simulations (4). This rigorous preparation ensures that the protein structure is in an optimal state for computational drug discovery efforts, particularly for the identification of novel CDK4/6 inhibitors from marine natural products.

**2.1.2 Active site.** The catalytic domain and binding site residues (ILE19, TYR24, ALA41, GLU99, GLN103, ASP104, and ALA162) of the CDK4/6 complex were elucidated through a comprehensive analysis of previously published literature [31–33], (PDB identifier: 5L2S). This initial characterization was further corroborated and refined using advanced computational tools CASTp [34] as well as AADS [35] to identify and characterize potential ligand-binding pockets on the protein surface based on its three-dimensional structure. This algorithm utilizes analytical methods to calculate the precise geometric properties of protein surface concavities and estimate the volume of each potential binding site. This dual approach of literature-based knowledge and computational prediction ensures a robust and accurate identification of the CDK4/6 active site.

**2.1.3 Structure-based virtual screening (SBVS).** The CMNPD database [36], and MNP library (http://docking.umh.es/downloaddb, (accessed on 13 December 2020) [37] were used for virtual screening to identify hit compounds. Duplicates were removed using Open Babel [38] and converted into PDBQT format. The SBVS was conducted using AutoDock Vina [39] and centered at coordinates center_x = 21.1675, center_y = 40.2099, center_z = -9.78291. The grid box was configured to target the protein's ligand-binding site. The optimal box size was calculated using eBoxSize Script [40], with a threshold of -7.0 kcal/mol. The compounds met the criteria and were selected for further research.

### 2.2 Drug-likeness and PAINS alert

To evaluate the drug-likeness and potential Drug-likeness properties of hit compounds identified through virtual screening RDkit [41] and Pandas [42] used. The initial evaluation was based on Lipinski's Rule of Five [43], Veber rule [44], Egan rule [45], Muegge rule [46], Ghose rule [47], Varma rule [48], GSK rule [49], Pfizer rule [50], and no PAINS alert [51] which correlates molecular properties with oral bioavailability in humans. Compounds that violated these criteria were excluded from further analysis, prioritizing compounds with higher success probabilities.

### 2.3 ADME

The attrition rate in drug development due to suboptimal absorption, distribution, metabolism, excretion, and toxicity (ADMET) properties is substantial, with approximately 40% of candidate compounds failing during preclinical stages due to inadequate biotransformation

and pharmacokinetics [52]. Moreover, unfavorable ADMET characteristics account for approximately 60% of failures during clinical trials [53]. Therefore, early prediction and optimization of these properties are critical in the drug discovery pipeline to mitigate late-stage attrition and associated costs. To address this challenge, we employed SwissADME [54], a robust web-based tool for predicting ADMET properties, to assess the pharmacokinetic profiles of our marine natural product-derived hit compounds. The computational workflow was optimized for high-throughput analysis by segmenting the input.smi files into 45 subsets, each containing 200 compounds, to accommodate the platform's upload limitations [55,56]. SwissADME output, consisting of multidimensional ADMET data, was merged and processed using a custom Python script for comprehensive analysis. The data underwent rigorous filtering based on key pharmacokinetic parameters, including intestinal absorption, blood-brain barrier permeability, aqueous solubility, logP, P-glycoprotein substrate specificity, and CYP1A2 inhibition potential. Suboptimal compounds were eliminated, prioritizing compounds with favorable pharmacokinetic properties.

### 2.4 Toxicity

Toxicity is a significant factor in drug development, accounting for 30% of lead candidate attrition. *In-silico* toxicity prediction is crucial for early-stage drug discovery. OSIRIS DataWarrior [57] is used to predict key toxicological parameters for fungal metabolite-derived compounds, providing quantitative estimates of toxicity risks and prioritizing compounds with favorable safety profiles early in the development process.

### 2.5 Consensus molecular docking

The study used a consensus molecular docking approach to improve the reliability and robustness of virtual screening results. The approach leverages the strengths of multiple docking algorithms to mitigate software biases and provide a more comprehensive assessment of ligand-protein interactions. We employed seven state-of-the-art molecular docking programs with distinct scoring functions and search algorithms. These programs included AutoDock-Tools 4.2 [58], idock [59], LeDock [60], Qvina 2 [61], Smina [62], AutoDock Vina 1.2.0 [63], PLANTS [64], and rDock [65]. To get a balanced assessment of a compound's performance across multiple docking algorithms Rank-by-Rank (RbR) method was used to calculate the final rank of each molecule by averaging its individual ranks across all docking programs. The goal was to obtain a more robust ranking of potential CDK4/6 inhibitors derived from marine natural products. This consensus approach helps account for individual docking algorithms' inherent limitations and biases, potentially leading to more reliable predictions of binding modes and affinities.

### 2.6 Molecular dynamics simulation

To elucidate the dynamic behavior and stability of the optimally docked CDK4/6-ligand complexes, we conducted comprehensive 500 nanosecond all-atom MD simulations using GROMACS [66]. The simulation protocol followed the methodology established in our previous studies [67–69], ensuring consistency and comparability of results. Each CDK4/6-ligand complex was subjected to a 500-nanosecond (ns) MD simulation under physiologically relevant conditions. The simulations were performed using the CHARMM36 all-atom force field (July 2022) [70,71] force field for proteins (3) and CHARMM force field parameter using CGenFF server [72,73] for ligands, with explicit TIP3P water model in a periodic boundary condition [74–76]. The study analyzed key parameters such as RMSD, RMSF, Rg, SASA, Hydrogen Bond Analysis, PCA, Protein-Ligand Interaction Timeline, and mm-PBSA to evaluate the

stability and conformational dynamics of protein-ligand complexes during a simulation. These parameters help quantify structural deviation, identify high-flexibility regions, evaluate protein structure compactness, assess changes in protein surface exposure, and estimate binding free energies. These analyses were performed using built-in GROMACS tools, MDAnalysis [77], NGLview [78] and ProLIF [79], providing a multifaceted view of the CDK4/6-ligand complex stability in a dynamic environment.

### 2.7 MTT assay

The human breast adenocarcinoma, MCF-7 cells were procured from NCCS, Pune, India, and cultured in DMEM media supplemented with 10% FBS, 1% penicillin-streptomycin, and 1% L-glutamine, maintained at 37°C in a 5% $CO_2$ humidified atmosphere. Cells were seeded in 96-well plates at $2 \times 10^3$ cells/well and allowed to adhere overnight. Stock solutions of CMNPD14217, CMNPD2744, CMNPD11585, and Ribociclib were prepared in DMSO and serially diluted in DMEM to final concentrations ranging from 12.5 μM to 0.024414 μM. Cells were treated with the test compounds, and controls (Ribociclib as standard) for 48 hours, with each concentration tested in triplicate. Following treatment, 50 μL of MTT solution (5 mg/mL in PBS) was added to each well and incubated for 4 hours at 37°C in the dark. The medium was removed, and 100 μL of DMSO was added to solubilize the formazan crystals. Absorbance was measured at 570 nm using a microplate reader. Cell viability was calculated relative to the vehicle control, $IC_{50}$ values were determined using nonlinear regression analysis, and the data was expressed as mean ± SD.

## 3. Results

### 3.1 Identification of hits by virtual screening

A structure-based virtual screening process identified 9497 distinct hit compounds with binding affinity less than -7.99 kcal/mol to the receptor, which was chosen for future investigation after removing duplicate compounds.

### 3.2 Drug-likeness and PAINS

We have applied multiple medicinal chemistry-based filters to identify and eliminate compounds with PAINS alerts. Fig 1 visualizes the pass/fail status of 9,497 compounds across various medicinal chemistry filters. Each row represents a different filter, including Lipinski's Rule of Five, Veber's criteria, Egan's criteria, Muegge's rules, Ghose's rules, Varma's rules, the GSK 4/400 rule, Pfizer's 3/75 rule, and the PAINS filter. This heatmap helps quickly identify which compounds pass or fail multiple filters, providing a visual summary of the drug-likeness evaluation.

Fig 2 represents the multi-tiered filtering process of the 9,497 compounds and shows the number of compounds that pass each successive filter, which visually represents the reduction in the number of compounds as more stringent filters are applied, providing a clear overview of the filtering process and the number of compounds that pass each stage. This approach provides a robust assessment of the compound library's potential for oral bioavailability, metabolic stability, and reduced likelihood of promiscuous binding, offering a refined subset of compounds with promising drug-like properties for further ADME study.

### 3.3 ADME

The attrition of lead compounds in clinical studies is frequently attributed to suboptimal ADME properties [80–82]. To mitigate this risk, silico pharmacokinetic profiles were predicted for each compound to eliminate those with unfavorable characteristics. From the initial

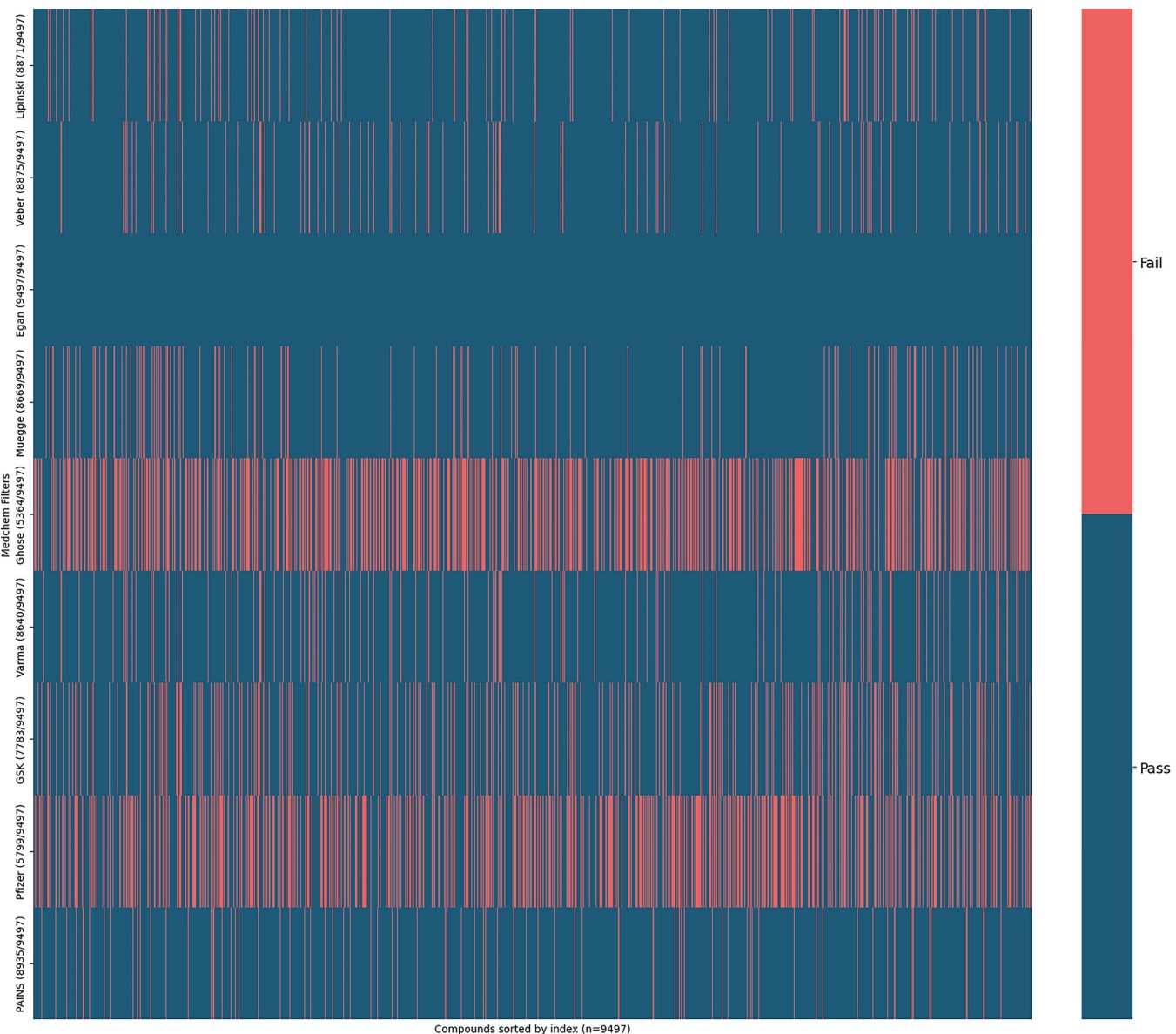

**Fig 1. Heatmap of drug-likeness and PAINS evaluation.**

2,344 compounds that passed drug-likeness and PAINS filters, only 50 met all of the following stringent ADME criteria: high aqueous solubility (classified as 'Soluble' or 'Very soluble' by ESOL, Ali, and Silicos-IT models), high gastrointestinal (GI) absorption, inability to permeate the blood-brain barrier (BBB), non-substrate for P-glycoprotein (Pgp), and non-inhibitor for key cytochrome P450 enzymes (CYP1A2, CYP2C19, CYP2C9, CYP2D6, and CYP3A4). This rigorous filtering process ensured the selection of compounds with promising pharmacokinetic profiles for further Toxicity studies.

### 3.4 Toxicity

Toxicity assessment is a critical component of the drug discovery pipeline, essential for ensuring safety and predicting potential adverse effects. We employed *in-silico* toxicity prediction

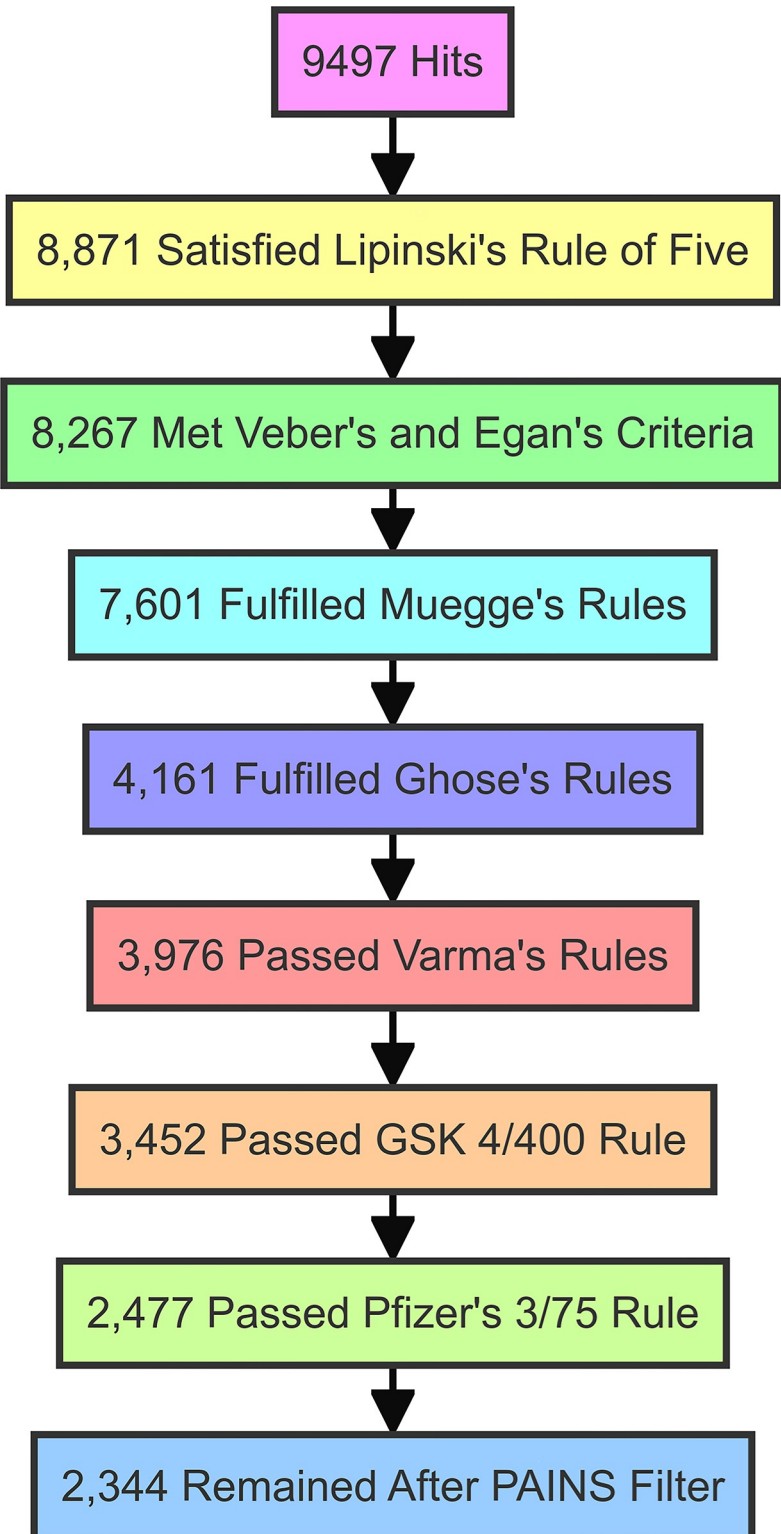

**Fig 2. The colorful funnel chart represents the multi-tiered filtering process of 9,497 compounds, showing the number of compounds that pass each successive filter stage.**

methods to evaluate the 50 compounds that passed previous pharmacokinetic filters. The assessment focused on key toxicological parameters including mutagenicity, tumorigenicity, reproductive toxicity, irritant potential, and structural alerts or undesirable moieties. Based on these criteria, 25 compounds were identified as having favorable toxicity profiles, demonstrating non-mutagenic properties, lack of tumorigenic potential, no predicted adverse effects on reproductive function, low irritant potential, and absence of known toxicophores or reactive functional groups. This rigorous toxicity screening process significantly reduced the risk of late-stage attrition due to safety concerns, allowing us to prioritize compounds with the most promising safety profiles for further investigation.

## 3.5 Consensus molecular docking

Consensus molecular docking has emerged as a robust approach to enhance the reliability and accuracy of virtual screening outcomes compared to single docking methods [83,84]. This technique facilitates the identification of optimal molecular conformations for binding to CDK4/6 with the lowest binding energy while improving pose prediction accuracy by integrating ligand-binding poses generated through diverse docking algorithms [82,85]. In our study, we employed a consensus docking approach using multiple docking programs to evaluate the binding affinity and modes of the 25 compounds that passed Toxicity filters. The results of the consensus molecular docking, including binding energies and consensus scores for each compound, are presented in Table 1.

**Table 1. Results of consensus molecular docking.**

| Compound ID | AutoDock Tools 4.2 | LeDock | Qvina | Smina | Vina | PLANTS SCORE | rDock Score |
|---|---|---|---|---|---|---|---|
| | Binding energy in kcal/mol | | | | | | |
| 194143-57-2 | -13.82 | -5.45 | -8.2 | -8.37179 | -8.26 | -79.8596 | -14.9023 |
| 205813-99-6 | -13.18 | -6.09 | -8.5 | -8.70485 | -8.599 | -90.4657 | -26.2504 |
| 207305-65-5 | -10.71 | -4.61 | -7.6 | -7.59515 | -7.814 | -48.0241 | -8.92162 |
| 223130-61-8 | -11.19 | -7.53 | -9.4 | -9.71733 | -9.455 | -73.2084 | -33.6306 |
| 251343-62-1 | -13.12 | -4.88 | -7.6 | -7.92827 | -7.54 | -74.1601 | -14.7578 |
| CMNPD10652 | -9.21 | -4.65 | -7.2 | -7.53275 | -7.067 | -79.239 | -11.9891 |
| CMNPD11585 | -12.43 | -7.35 | -10.6 | -10.6076 | -10.541 | -83.8021 | -36.2474 |
| CMNPD12908 | -9.11 | -3.99 | -6 | -6.43511 | -5.994 | -20.1983 | -37.0736 |
| CMNPD1346 | -10.67 | -5.93 | -8.8 | -9.50728 | -8.622 | -85.0765 | -23.3839 |
| CMNPD13472 | -12.45 | -5.83 | -9.3 | -9.32971 | -9.708 | -67.0398 | -35.665 |
| CMNPD14214 | -9.36 | -4.96 | -9 | -9.01676 | -9.228 | -60.5698 | -10.614 |
| CMNPD14217 | -12.39 | -5.61 | -10 | -10.2387 | -9.929 | -85.4462 | -34.8739 |
| CMNPD2018 | -11.63 | -4.12 | -7.2 | -7.54516 | -7.155 | -68.0614 | -8.30005 |
| CMNPD2744 | -10.83 | -7.32 | -8.9 | -9.50247 | -9.029 | -93.0371 | -21.8022 |
| CMNPD3195 | -12.58 | -3.84 | -7 | -7.56716 | -6.801 | -71.031 | -11.6868 |
| CMNPD4645 | -11.32 | -5.98 | -7.6 | -7.99042 | -7.673 | -90.7608 | -21.0265 |
| CMNPD4920 | -12.14 | -5.71 | -8.4 | -8.58632 | -8.384 | -83.9795 | -16.0704 |
| CMNPD5678 | -11.66 | -5.06 | -8.7 | -9.20217 | -8.378 | -75.2528 | -11.6789 |
| CMNPD5679 | -11.7 | -4.07 | -5.5 | -5.90156 | -5.392 | -76.619 | -1.19659 |
| CMNPD5682 | -11.83 | -4.51 | -6.9 | -7.06105 | -6.952 | -71.4684 | -13.2668 |
| CMNPD646 | -10.78 | -4.99 | -6.8 | -6.97034 | -6.773 | -73.5001 | -11.1624 |
| CMNPD7036 | -12.42 | -4.33 | -7.6 | -8.04135 | -7.646 | -80.1545 | -45.955 |
| CMNPD7986 | -15.4 | -8.03 | -9.1 | -9.40776 | -9.364 | -128.572 | -48.8013 |
| CMNPD8752 | -12.11 | -5.33 | -7.7 | -7.83148 | -7.881 | -78.9698 | -10.1984 |
| CMNPD9390 | -9.11 | -4.57 | -7.6 | -7.56806 | -7.593 | -49.3505 | -9.30684 |

**Table 2. Results of consensus molecular docking after ranking them.**

| Compound ID | ADT Ranking | LeDock Ranking | Qvina Ranking | Smina Ranking | Vina Ranking | PLANTS Ranking | rDock Ranking |
|---|---|---|---|---|---|---|---|
| 194143-57-2 | 2 | 11 | 12 | 12 | 12 | 10 | 13 |
| 205813-99-6 | 3 | 5 | 10 | 10 | 9 | 4 | 8 |
| 207305-65-5 | 20 | 18 | 14 | 17 | 14 | 24 | 23 |
| 223130-61-8 | 17 | 2 | 3 | 3 | 4 | 17 | 7 |
| 251343-62-1 | 4 | 16 | 14 | 15 | 18 | 15 | 14 |
| CMNPD10652 | 23 | 17 | 19 | 21 | 20 | 11 | 16 |
| CMNPD11585 | 7 | 3 | 1 | 1 | 1 | 8 | 4 |
| CMNPD12908 | 24 | 24 | 24 | 24 | 24 | 25 | 3 |
| CMNPD1346 | 21 | 7 | 8 | 4 | 8 | 6 | 9 |
| CMNPD13472 | 6 | 8 | 4 | 7 | 3 | 21 | 5 |
| CMNPD14214 | 22 | 15 | 6 | 9 | 6 | 22 | 20 |
| CMNPD14217 | 9 | 10 | 2 | 2 | 2 | 5 | 6 |
| CMNPD2018 | 15 | 22 | 19 | 20 | 19 | 20 | 24 |
| CMNPD2744 | 18 | 4 | 7 | 5 | 7 | 2 | 10 |
| CMNPD3195 | 5 | 25 | 21 | 19 | 22 | 19 | 17 |
| CMNPD4645 | 16 | 6 | 14 | 14 | 15 | 3 | 11 |
| CMNPD4920 | 10 | 9 | 11 | 11 | 10 | 7 | 12 |
| CMNPD5678 | 14 | 13 | 9 | 8 | 11 | 14 | 18 |
| CMNPD5679 | 13 | 23 | 25 | 25 | 25 | 13 | 25 |
| CMNPD5682 | 12 | 20 | 22 | 22 | 21 | 18 | 15 |
| CMNPD646 | 19 | 14 | 23 | 23 | 23 | 16 | 19 |
| CMNPD7036 | 8 | 21 | 14 | 13 | 16 | 9 | 2 |
| CMNPD7986 | 1 | 1 | 5 | 6 | 5 | 1 | 1 |
| CMNPD8752 | 11 | 12 | 13 | 16 | 13 | 12 | 21 |
| CMNPD9390 | 24 | 19 | 14 | 18 | 17 | 23 | 22 |

Following consensus molecular docking, we implemented the Rank-by-Rank (RbR) strategy to further optimize and prioritize our results [86]. This method involves ranking the compounds based on their performance across multiple scoring functions, with the most favorable outcome receiving a rank of 1 shown in Table 2. The RbR approach compares a compound's relative performance across different scoring functions, mitigating biases inherent to individual methods [87]. A final unified ranking is then assigned to each compound by calculating the average of its ranks across all scoring functions. This process allows for a more holistic evaluation of binding affinity predictions, accounting for the strengths and limitations of various scoring algorithms [88]. Compounds achieving the best average rank across methods are identified as the most promising candidates for further investigation, including experimental validation and structural optimization. This systematic ranking approach enhances the robustness of our virtual screening pipeline and increases confidence in the selection of lead compounds for CDK4/6 inhibition.

The radar plot (Fig 3) is employed to illustrate the consensus docking findings, illustrating the ranks of compounds across various procedures on a circular graph. To facilitate the identification of potential candidates, compounds with the highest rankings have been shown with a variety of colors.

The six compounds with the following identifiers (205813-99-6, 223130-61-8, CMNPD11585, CMNPD14217, CMNPD2744, CMNPD7986) have been selected for Molecular Dynamics Simulation because of their high likelihood to interact with CDK4/6. This is supported by their favorable free binding energy and docking program scores.

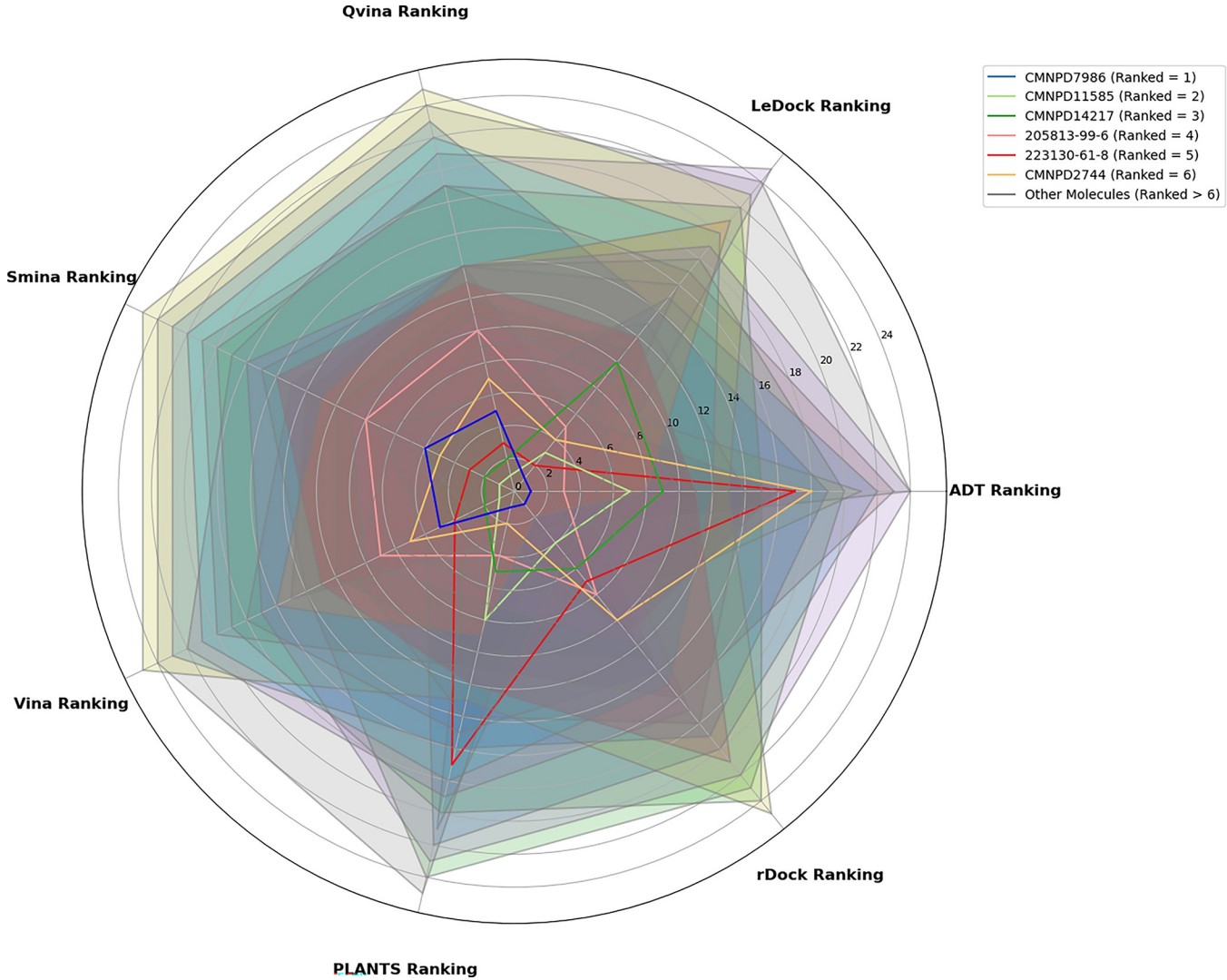

**Fig 3. Results of consensus molecular docking after RbR.**

## 3.6 Molecular dynamics simulation

A 500 nanosecond Molecular Dynamics Simulation was employed to assess the effects of ligand binding on protein stability, flexibility, interaction patterns, and energetics. This analysis yielded essential information for optimizing drug design and efficacy.

**3.6.1 RMSD.** The RMSD trajectories of six ligand-bound protein complexes, along with the protein alone, were analyzed to evaluate conformational stability over a 500-nanosecond molecular dynamics simulation shown in Fig 4.

The protein exhibited an average RMSD of 3.129 Å ± 0.299 Å, serving as a baseline for comparison. Among the complexes, CMNPD2744 demonstrated the most stable interaction with the lowest average RMSD of 2.806 Å ± 0.205 Å, maintaining consistent stability throughout the simulation. CMNPD14217 also showed promising stability with an average RMSD of 3.190 Å ± 0.339 Å. Complexes 223130-61-8 and CMNPD7986 exhibited moderate stability with average RMSD values of 3.259 Å ± 0.204 Å and 3.261 Å ± 0.300 Å, respectively, indicating a balance between stability and flexibility. Conversely, 205813-99-6 and CMNPD11585

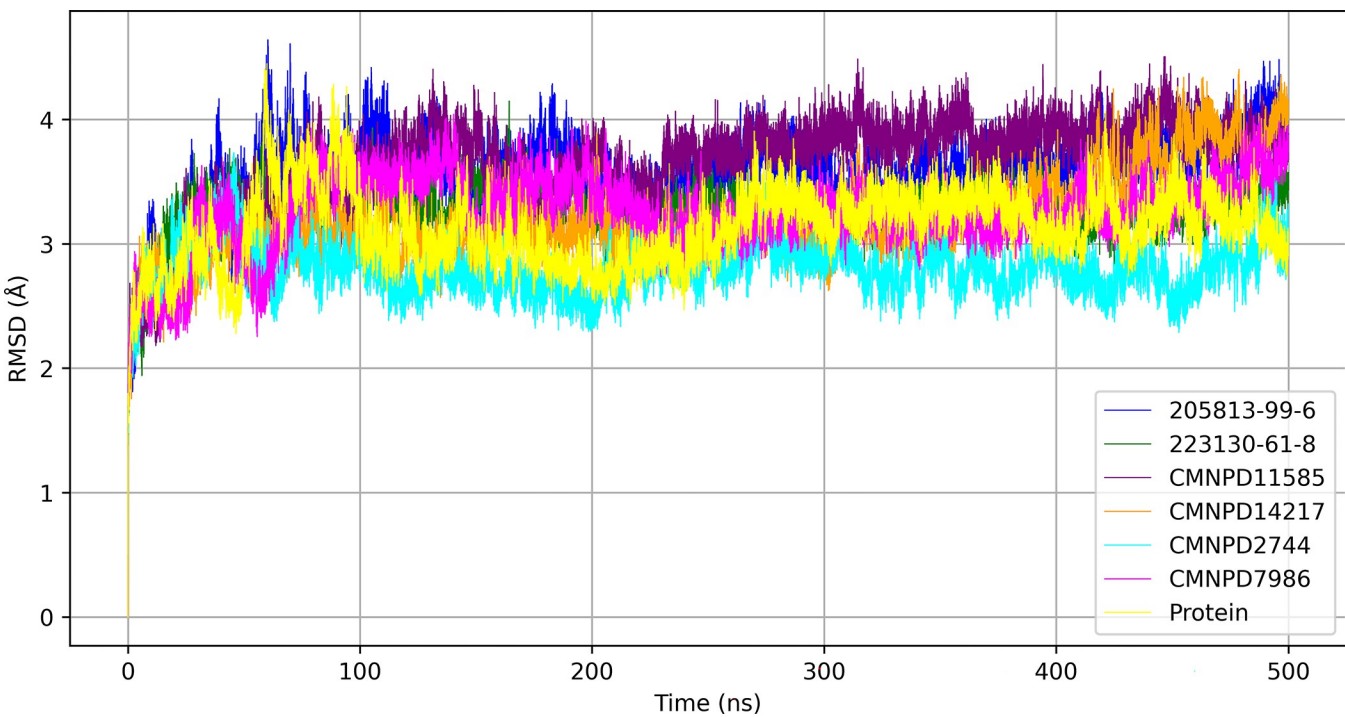

**Fig 4. RMSD trajectories of protein and different ligand-bound protein over time highlighting stability and conformational dynamics.**

displayed higher average RMSD values of 3.562 Å ± 0.316 Å and 3.658 Å ± 0.359 Å, respectively, suggesting greater conformational flexibility. Notably, these complexes showed pronounced RMSD spikes, particularly in the 50–150 ns range, indicating episodes of reduced stability due to transient structural reconfigurations. CMNPD7986 presented a gradual RMSD increase, implying a slow drift from the initial conformation, potentially affecting long-term stability. The consistently low RMSD of CMNPD2744 suggests it may form the most stable protein-ligand complex, indicating stronger binding affinity. In contrast, the higher RMSD values and fluctuations observed for 205813-99-6 and CMNPD11585 suggest more dynamic interactions with the protein's binding site, potentially impacting their efficacy as drug candidates.

**3.6.2 RMSF.** The Root Mean Square Fluctuation (RMSF) trajectories of six ligand-bound protein complexes, along with the protein alone, were analyzed to evaluate residue flexibility over a 500 ns span in molecular dynamics simulations shown in Fig 5.

The protein exhibited an average RMSF of 1.499 Å ± 1.065 Å, providing a baseline for comparison. The CMNPD2744 demonstrated the lowest average RMSF of 1.355 Å ± 0.780 Å, indicating the most consistent residue stability across the protein structure when bound to this ligand. 223130-61-8 also showed promising stability with a low average RMSF of 1.438 Å ± 0.909 Å, suggesting relatively rigid protein conformation. CMNPD11585 exhibited moderate flexibility with an average RMSF of 1.534 Å ± 1.037 Å, balancing between stability and necessary protein dynamics. CMNPD14217 and 205813-99-6 displayed higher average RMSF values of 1.738 Å ± 1.090 Å and 1.756 Å ± 1.112 Å respectively, indicating greater overall residue flexibility when bound to these ligands. Notably, all ligands showed distinct RMSF spikes around residue numbers 50, 100, and 170–180, with the most pronounced peak occurring around residue 170–180. This suggests the presence of highly flexible regions or loops in the protein structure, which may play crucial roles in ligand binding or protein function. CMNPD2744 and

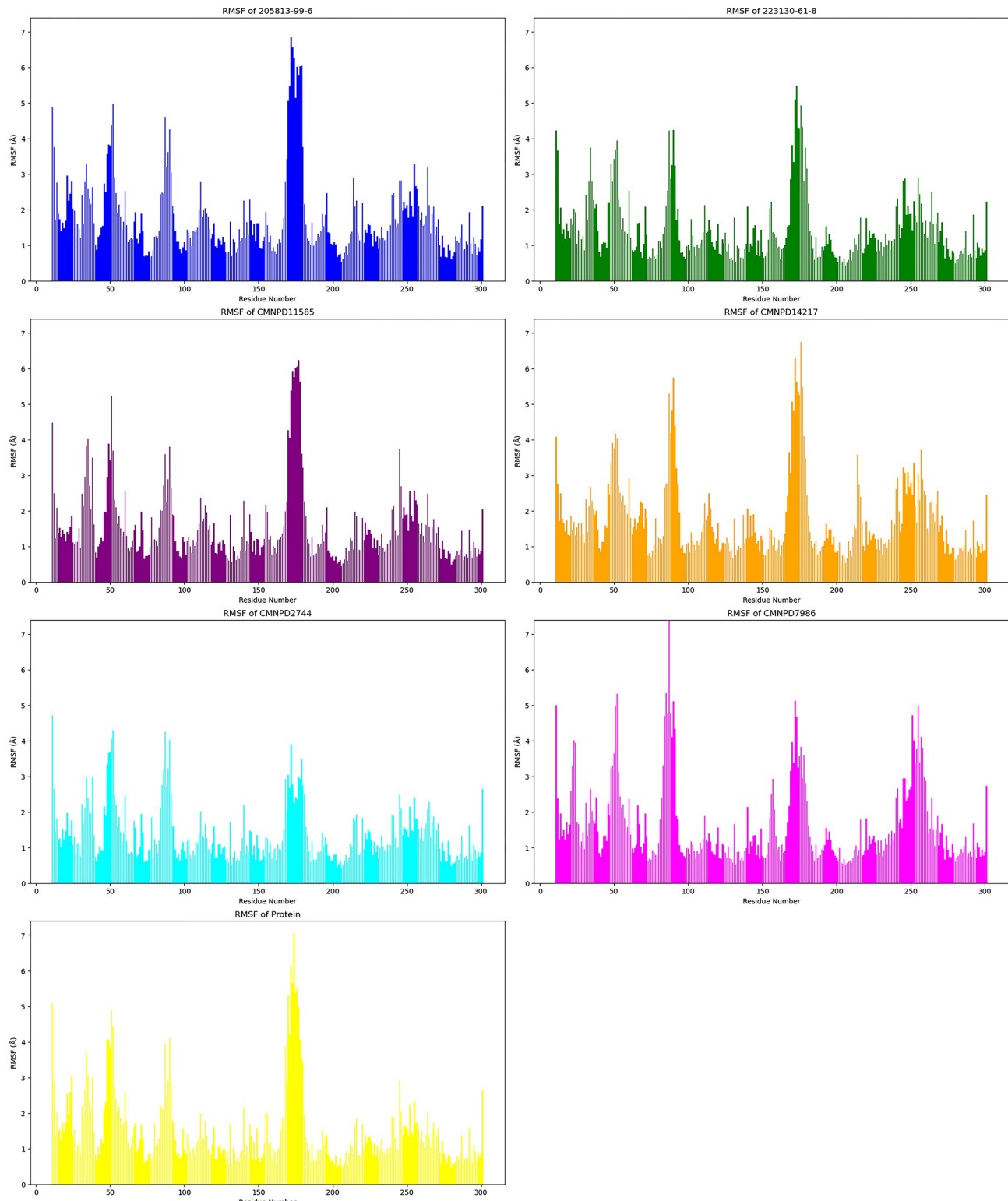

**Fig 5. Localized flexibility analysis of ligand-bound proteins via RMSF.**

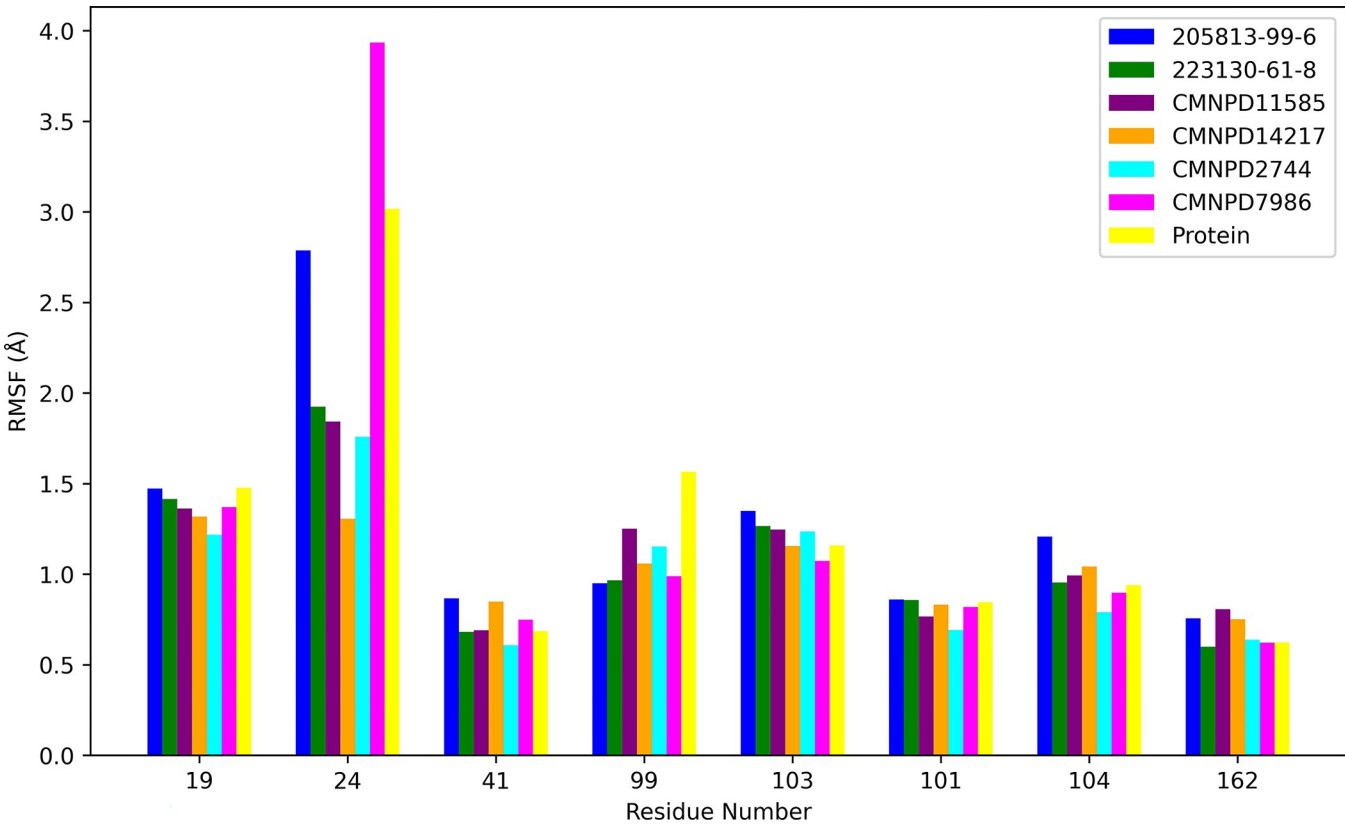

**Fig 6. RMSF highlights differential flexibility in key protein domains across various ligand-protein complexes.**

223130-61-8 maintained lower RMSF values even in these flexible regions, implying they might stabilize the protein structure more effectively. In contrast, 205813-99-6 and CMNPD14217 showed higher RMSF peaks, particularly around residue 170–180, indicating they may induce more significant local conformational changes in these flexible regions. The consistently low RMSF of CMNPD2744 suggests it may form the most stable protein-ligand complex among the tested compounds, potentially indicating stronger binding affinity and less perturbation of the overall protein structure. Conversely, the higher RMSF values observed for 205813-99-6 and CMNPD14217 suggest these ligands may induce more dynamic protein behavior, which could impact their efficacy as potential drug candidates.

The RMSF analysis of key active site residues (Fig 6) provides crucial insights into how different ligands interact with and influence the protein's binding pocket. The residues analyzed include ILE19, TYR24, ALA41, GLU99, GLN103, ASP104, and ALA162. CMNPD2744 and 223130-61-8 generally induce lower RMSF values across most residues, suggesting they form more stable complexes with the protein. This stability could indicate stronger binding affinities, making them potentially more effective drug candidates. In contrast, CMNPD7986 causes an exceptionally high fluctuation in residue TYR24 (RMSF nearly 4 Å), pointing to a unique binding mode that significantly perturbs this part of the active site. This dramatic effect could be beneficial if it induces a favorable conformational change or detrimental if it disrupts critical protein-ligand interactions. 205813-99-6 tends to induce slightly higher fluctuations across several residues, particularly noticeable in ILE19, TYR24, and ASP104. This pattern suggests a more dynamic binding mode, which could allow for greater adaptability in the binding pocket but might come at the cost of overall complex stability. CMNPD11585 and CMNPD14217

appear to strike a balance, maintaining relatively low RMSF values for most residues while allowing for some flexibility, particularly evident in their moderate effect on TYR24. Interestingly, all ligands maintain low RMSF values for residues ALA41, VAL101, and ALA162, indicating these amino acids likely play a crucial role in anchoring the ligands within the binding site. The consistent stability of these residues across all ligands suggests they could be key targets for maintaining essential protein-ligand interactions in future drug design efforts. The variable effects of the ligands on TYR24 are particularly noteworthy. While CMNPD7986 causes extreme fluctuation, others like CMNPD11585 and CMNPD14217 induce much less movement. This variability implies that TYR24 might be a critical determinant of ligand specificity and binding mode, potentially serving as a flexible 'gatekeeper' in the active site.

**3.6.3 Radius of gyration.** To assess the compactness and structural integrity of protein-ligand complexes throughout the molecular dynamics simulation, the radius of gyration (Rg) was calculated and is shown in Fig 7.

The protein alone exhibited an average Rg of 20.190 Å ± 0.155 Å, providing a reference point for comparison. The ligand 205813-99-6 bound protein displayed the highest average Rg of 20.794 Å ± 0.236 Å, indicating a less compact structure and potentially greater flexibility in the complex. In contrast, the 223130-61-8 bound protein exhibited the lowest average Rg value of 20.027 Å ± 0.147 Å, suggesting a more tightly packed conformation and stable complex throughout the simulation. The CMNPD11585 bound protein presented an intermediate average Rg of 20.423 Å ± 0.205 Å, showing a protein structure that generally maintains moderate compactness with some conformational variations. The CMNPD2744 bound protein demonstrated a stable complex with an average Rg of 20.394 Å ± 0.134 Å, indicating consistent compactness and minimal structural fluctuations during the simulation. The remaining ligands, CMNPD14217 and CMNPD7986, showed similar levels of compactness with average Rg values of 20.465 Å ± 0.196 Å and 20.338 Å ± 0.158 Å, respectively, suggesting stable complexes with minor conformational shifts throughout the simulation period.

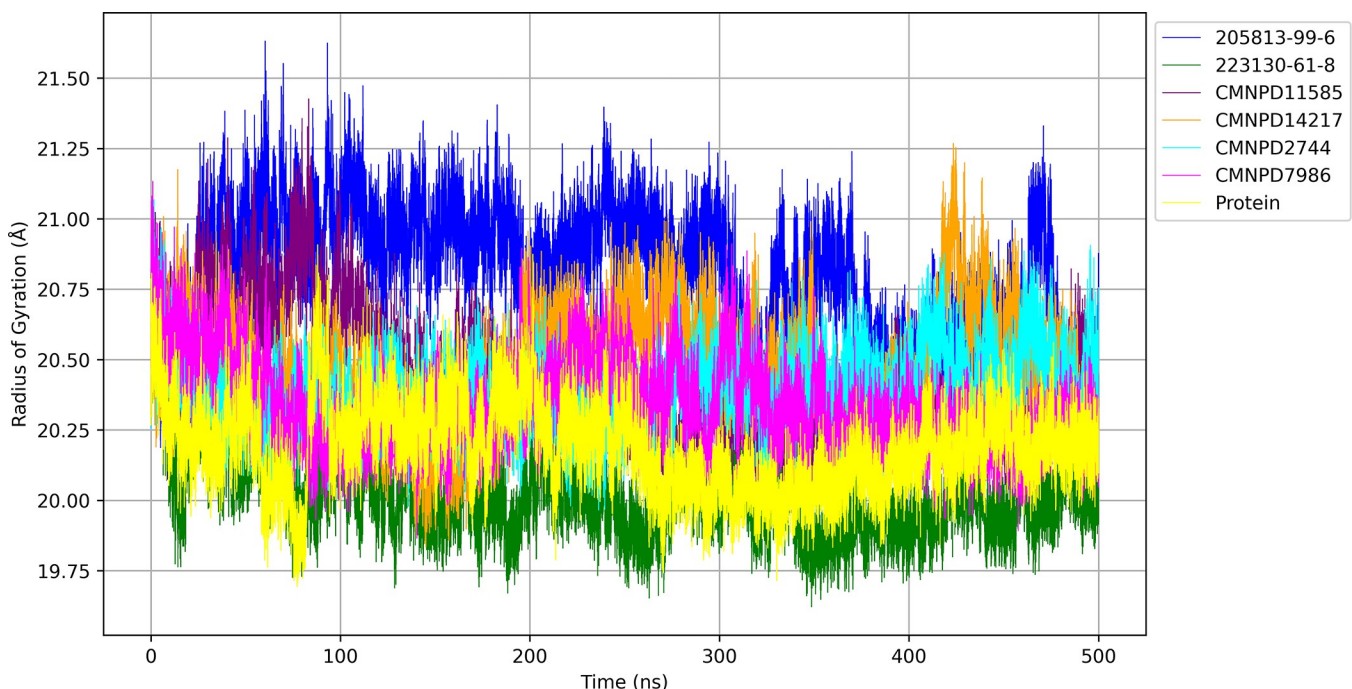

**Fig 7. Rg over time.**

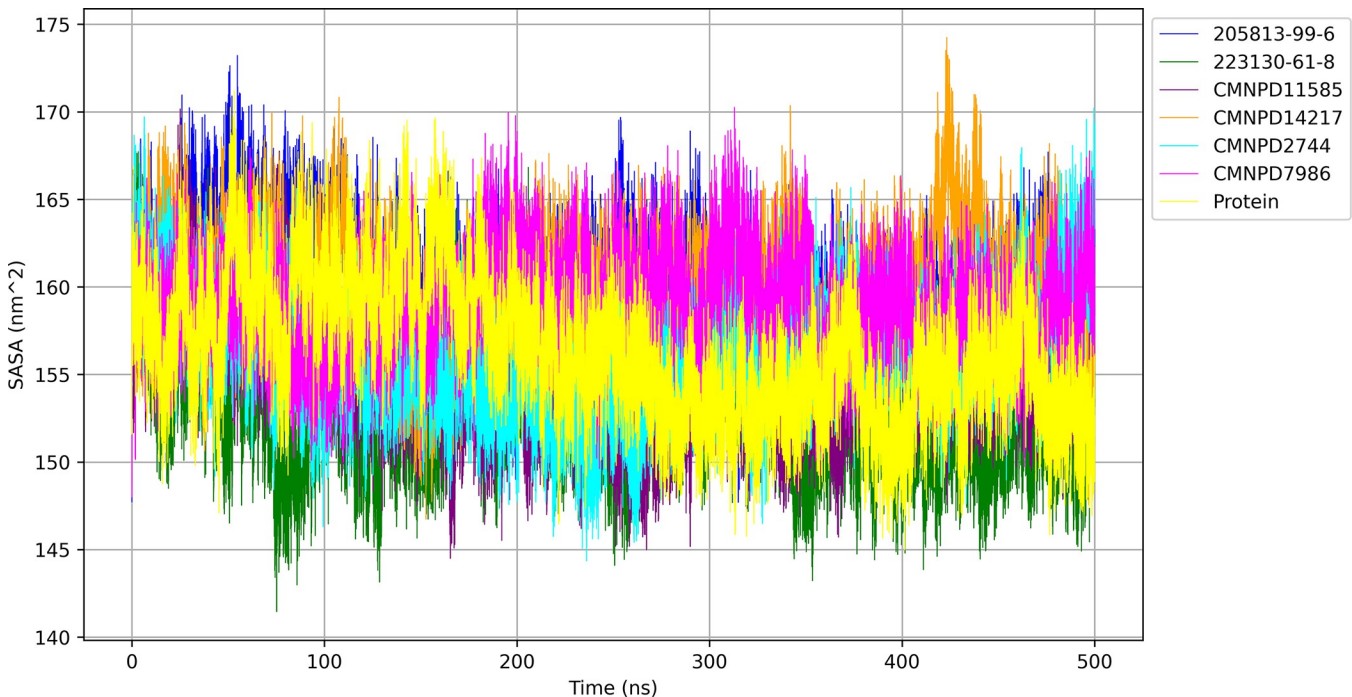

**Fig 8. SASA of protein and protein-ligand complexes over 500 ns.**

**3.6.4 Solvent accessible surface area.** To evaluate the exposure of protein-ligand complexes to the surrounding solvent during molecular dynamics simulations, the solvent-accessible surface area (SASA) was calculated and is shown in Fig 8.

The protein alone exhibited an average SASA of 156.351 nm$^2$ ± 3.830 nm$^2$, serving as a reference for comparison. The CMNPD14217 bound protein exhibited the highest average SASA of 159.764 nm$^2$ ± 3.151 nm$^2$, indicating the greatest overall surface exposure to solvent among the complexes studied. Closely following, the 205813-99-6 bound protein displayed a high average SASA of 159.264 nm$^2$ ± 3.255 nm$^2$, suggesting a similarly expansive solvent-accessible surface. The CMNPD7986 bound protein presented an intermediate average SASA of 158.933 nm$^2$ ± 2.784 nm$^2$, demonstrating moderate solvent exposure with the lowest fluctuations among all complexes. CMNPD2744 and CMNPD11585 bound proteins showed comparable levels of solvent accessibility with average SASA values of 155.992 nm$^2$ ± 3.222 nm$^2$ and 155.254 nm$^2$ ± 3.265 nm$^2$, respectively, indicating well-maintained surface exposure throughout the simulation. Notably, the 223130-61-8 bound protein demonstrated the lowest average SASA of 152.968 nm$^2$ ± 3.150 nm$^2$, suggesting a more compact structure with reduced solvent-accessible surface area compared to the other complexes. These SASA values provide insights into the potential for solvent interactions and the overall conformational behavior of each protein-ligand complex during the simulation period.

**3.6.5 Principle component analysis.** The principal component analysis (PCA) plot provides valuable insights into the conformational space explored by each protein-ligand complex during the molecular dynamics simulation. This 2D projection of the trajectory onto the first two principal components (PC1 and PC2) shown in Fig 9 reveals distinct patterns of conformational sampling for each ligand.

The Principal Component Analysis (PCA) of the ligand-bound protein complexes reveals diverse conformational dynamics across the compounds. CMNPD14217 demonstrates the

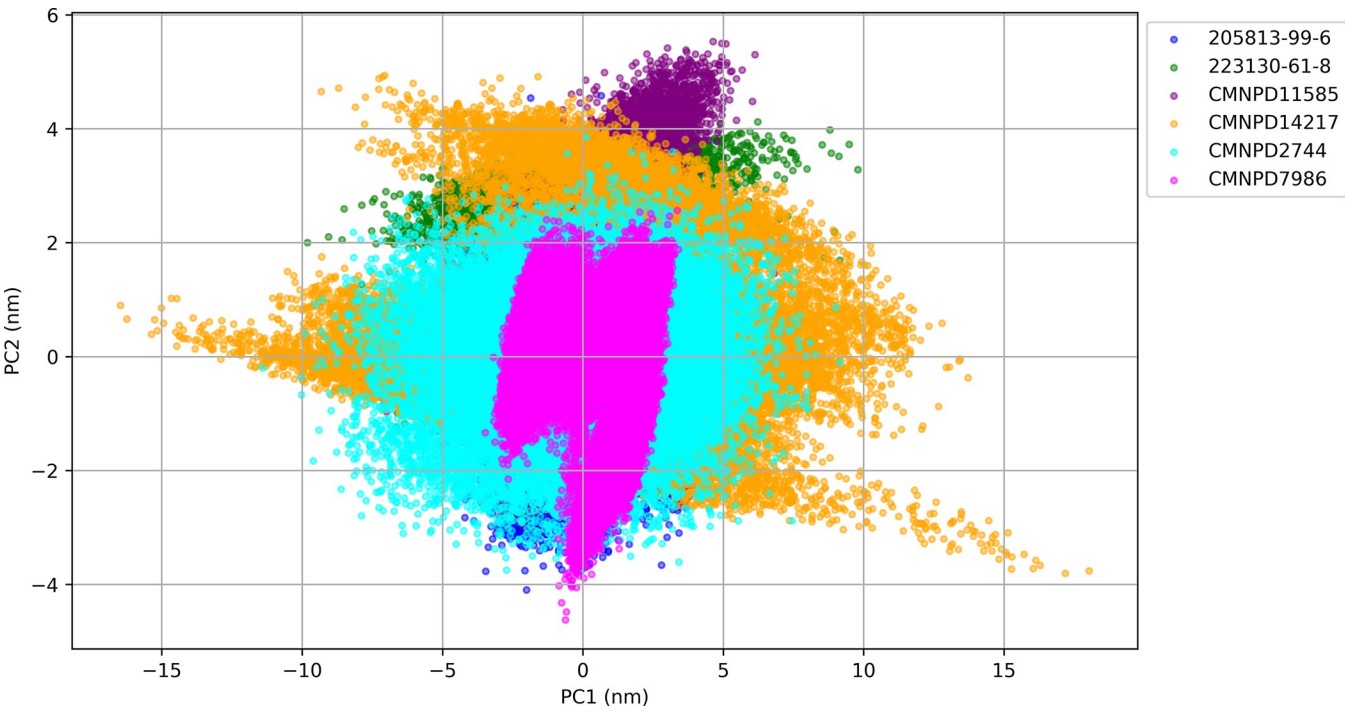

**Fig 9. Principal component analysis for ligand-protein complexes.**

most extensive flexibility with the largest PC1 range (-16.472 to 18.030 nm), suggesting a highly adaptable binding mode. In contrast, CMNPD7986 exhibits the most compact distribution in the PCA plot with the smallest PC1 range (-3.475 to 3.402 nm), indicating a highly stable and rigid complex. 223130-61-8 shows significant flexibility along PC1 (-9.814 to 9.790 nm) while forming a distinct cluster, suggesting a specific conformational preference despite its flexibility. CMNPD2744 displays a large PC1 range (-11.408 to 9.155 nm) but a narrow PC2 range, indicating anisotropic flexibility. CMNPD11585 presents moderate ranges for both PC1 (-7.684 to 6.821 nm) and PC2 (-3.007 to 5.531 nm), with a compact distribution suggesting stable conformational states. Interestingly, 205813-99-6 shows the smallest overall ranges (PC1: -4.926 to 4.168 nm, PC2: -4.095 to 4.580 nm) but a diffuse distribution in the PCA plot, implying varied but limited motions. These diverse conformational behaviors offer valuable insights for drug discovery, highlighting compounds suitable for different binding mechanisms: CMNPD14217 for induced-fit scenarios, CMNPD7986 for targets requiring rigid ligands, 223130-61-8 for stabilizing specific protein conformations, and 205813-99-6 as a starting point for optimization due to its limited but varied flexibility. This PCA data provides a foundation for tailoring drug design strategies to specific target proteins and desired mechanisms of action.

**3.6.6 Free energy landscape.** The free energy landscape analysis was conducted to evaluate the conformational stability and energy minima of the ligand-bound protein complexes. Fig 10 provides crucial insights into the most stable conformations that the protein-ligand complexes can adopt and the energy barriers between these states. The contour maps and 3D representations of the free energy minima offer a comprehensive visual representation of the energetic landscape. Each energy minimum represents a conformation where the protein-ligand complex is most stable, with the energy barriers indicating the stability and the likelihood of transitioning between different conformations.

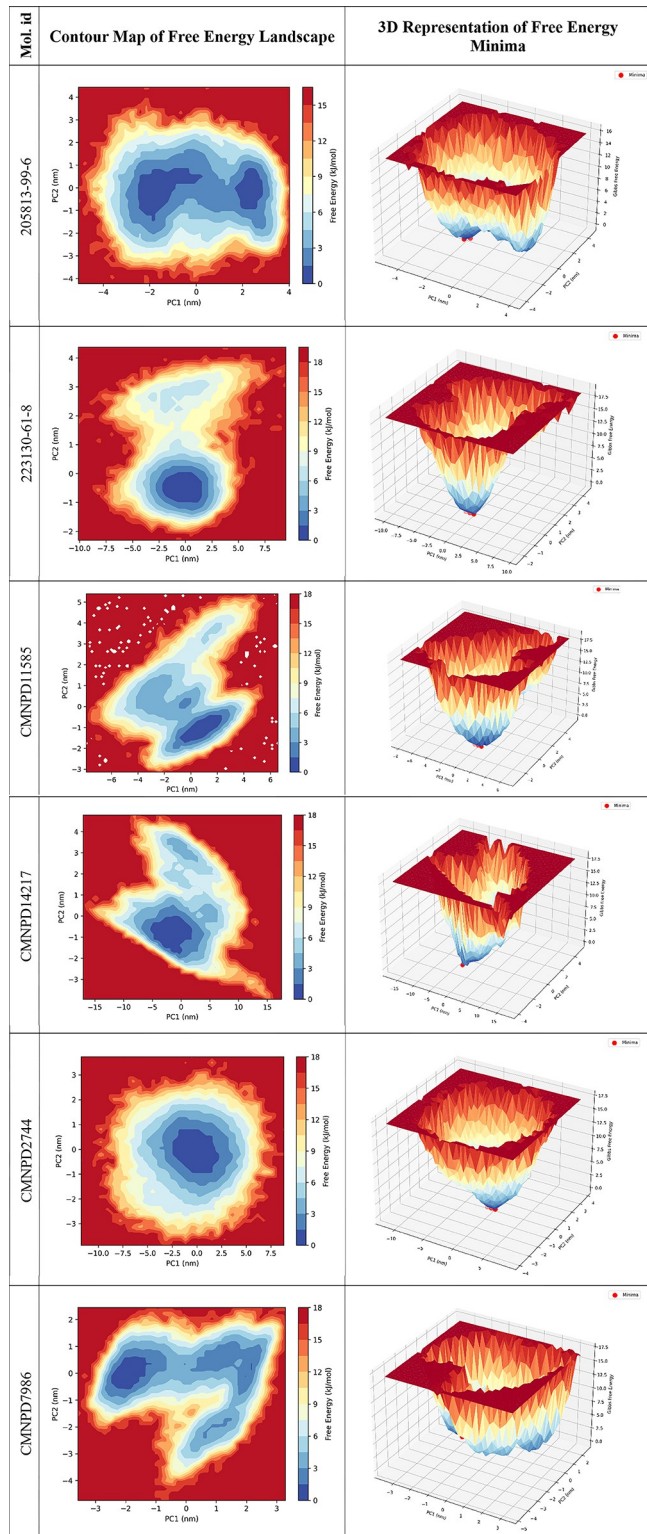

**Fig 10. Thermodynamic profiling of ligand-protein interactions through free energy landscape analysis.**

The FEL revealed that the 205813-99-6-bound protein exhibited multiple shallow energy minima spread across a broad area of the conformational space. This suggests a highly flexible complex with numerous accessible states, indicating a dynamic binding interaction. In contrast, the 223130-61-8-bound protein displayed a more focused energy landscape with a deep, well-defined global minimum, highlighting a stable and specific binding conformation with potentially strong affinity. The CMNPD11585-bound protein presented an interesting landscape with two distinct energy basins of similar depth, suggesting two equally favorable binding modes. This characteristic could indicate the potential for dual-action mechanisms or allosteric modulation. The CMNPD14217-bound protein exhibited the most promising landscape, featuring a single, deep global minimum with a clear funnel-like shape. This implies not only strong and specific binding but also favorable binding kinetics. CMNPD2744 displayed a relatively flat landscape with multiple shallow minima, indicating a weak and potentially non-specific binding interaction. This suggests that CMNPD2744 might require significant optimization to improve its binding characteristics. Conversely, CMNPD7986 showed a more structured landscape with a distinct global minimum, suggesting stronger binding and more defined conformational preferences compared to CMNPD2744. These observations imply that ligands such as 223130-61-8 and CMNPD14217, which exhibit deep, well-defined energy minima, are likely to form more robust and stable protein-ligand complexes. This stability is crucial for effective therapeutic efficacy, as stable binding interactions often correlate with better drug performance. The unique dual-minima landscape of CMNPD11585 presents intriguing possibilities for advanced drug design strategies, potentially offering dual-action or allosteric modulation capabilities. Conversely, ligands like 205813-99-6, CMNPD2744, and CMNPD2744, with greater flexibility and fewer stable states, may require further optimization to enhance binding stability and specificity. However, their flexibility could be advantageous when targeting proteins with induced-fit binding mechanisms or when designing drugs for proteins with high conformational plasticity. Ligands that demonstrate deep, well-defined energy minima with clear binding funnels, such as CMNPD14217 and 223130-61-8, should be given high priority for further development. These compounds are more likely to exhibit the desired combination of strong affinity, high specificity, and favorable binding kinetics essential for successful drug candidates.

**3.6.7 Hydrogen bond formation.** The hydrogen bond formation analysis over the 500 ns simulation shown in Fig 11 reveals distinct interaction patterns for each ligand-protein complex.

The CMNPD7986 demonstrates the most stable and numerous hydrogen bonds, consistently maintaining 3–4 and often reaching 5–6 bonds, indicating a highly specific and strong binding mode. CMNPD2744 shows a similarly high number of hydrogen bonds but with more fluctuations, suggesting a strong yet dynamic interaction. CMNPD14217 exhibits a highly variable pattern, ranging from 0 to 5 bonds, aligning with its flexibility observed in the PCA. CMNPD11585 maintains a moderate number of hydrogen bonds (typically 2–3) with occasional spikes, indicating a stable interaction with some flexibility. 223130-61-8 forms fewer hydrogen bonds on average (1–2), suggesting a binding mode potentially reliant on other types of interactions. 205813-99-6 demonstrates the fewest and most inconsistent hydrogen bonds, fluctuating between 0 and 2, indicating a less specific or weaker binding mode. These observations provide crucial insights for drug discovery: CMNPD7986 emerges as a promising candidate due to its strong and consistent hydrogen bonding, while CMNPD2744 and CMNPD11585 also show favorable patterns. CMNPD14217's variable bonding could be advantageous for induced-fit mechanisms, whereas 223130-61-8 and 205813-99-6 might benefit from modifications to enhance their hydrogen bonding capabilities.

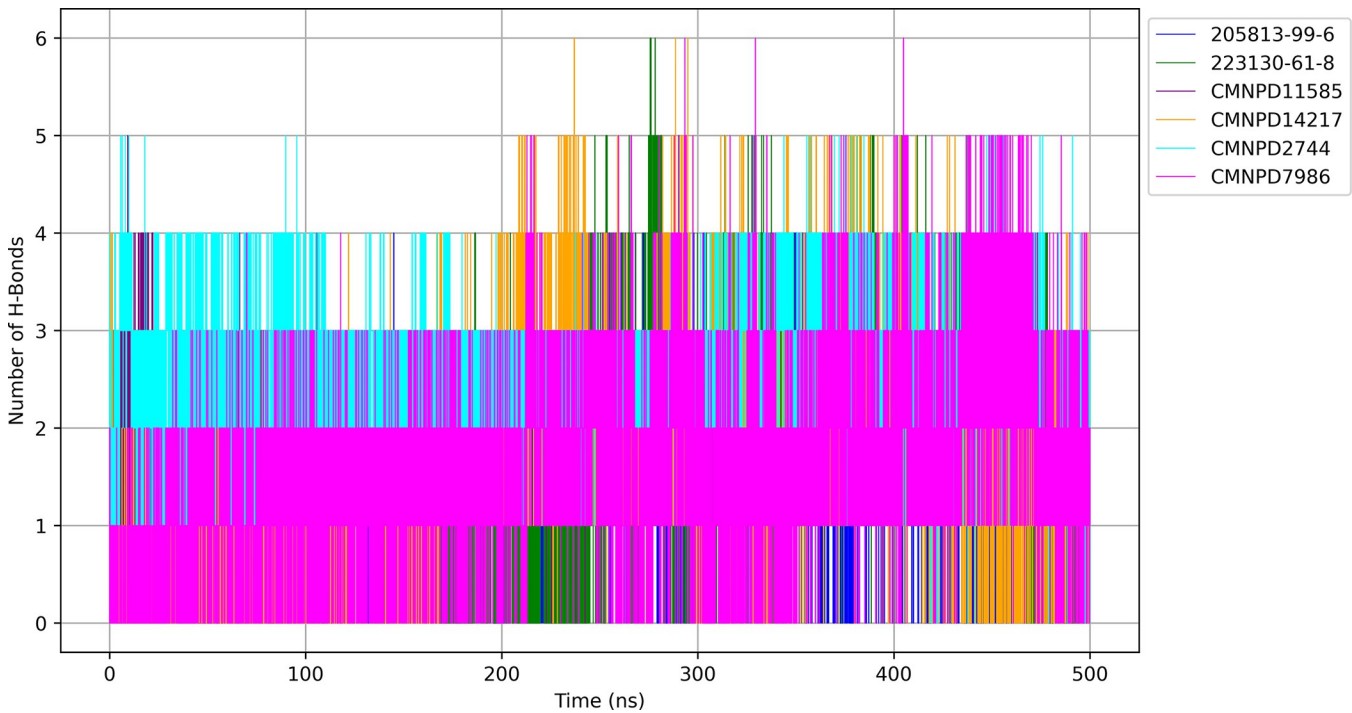

**Fig 11. Hydrogen bonding dynamics in ligand-protein complexes over simulation time.**

**3.6.8 Protein-ligands in interaction timeline.** Fig 12 depicts that the 205813-99-6 bound protein complex has both stable and transient interactions. Key residues such as VAL169, ASP153, ASP152, LEU142, ASP94, VAL91, ALA31, ILE9, and GLY10 showed persistent contacts, while a few other residues exhibited intermittent interactions.

While ligand 223130-61-8 (Fig 13) with the protein complex has numerous intermittent interactions, indicating potential instability in parts of the binding mode. The simulation shows gaps in interaction for several residues, which could signify temporary unbinding events or conformational changes affecting ligand position. The presence of various interaction types (hydrophobic, water-mediated, hydrogen bonds) suggests suboptimal binding affinity or specificity. While the ligand maintains contact throughout the simulation, the variability in interaction patterns possibly indicating issues with binding stability or off-target effects.

The molecular dynamics of ligand CMNPD11585 with the protein complex (Fig 14) reveals persistent interactions observed with key residues such as ASP153, ALA152, LEU142, ASN140, PHE88, ALA31, VAL17, and ILE9, indicating stable binding points. The ligand engages in diverse interaction types, including hydrophobic, water-mediated contacts, cationic interactions, and hydrogen bonds, suggesting a complex binding mechanism. However, some residues like GLN139, GLY10, and HIS90 show intermittent or weak interactions, potentially indicating areas of instability. Notably, GLU89 displays gaps in its interaction pattern, which could signify temporary loss of contact. While the overall binding appears stable, these inconsistencies suggest room for optimization to enhance binding affinity and stability. The diverse interaction profile could contribute to specificity but may also raise concerns about potential off-target effects.

Ligand CMNPD14217 with the protein complex (Fig 15) reveals key residues such as ASP153, ALA152, LEU142, VAL91, HIS90, GLU89, PHE88, LYS33, ALS31, VAL17, TYR14, and ILE9 exhibit strong, consistent interactions throughout the simulation, indicating stable

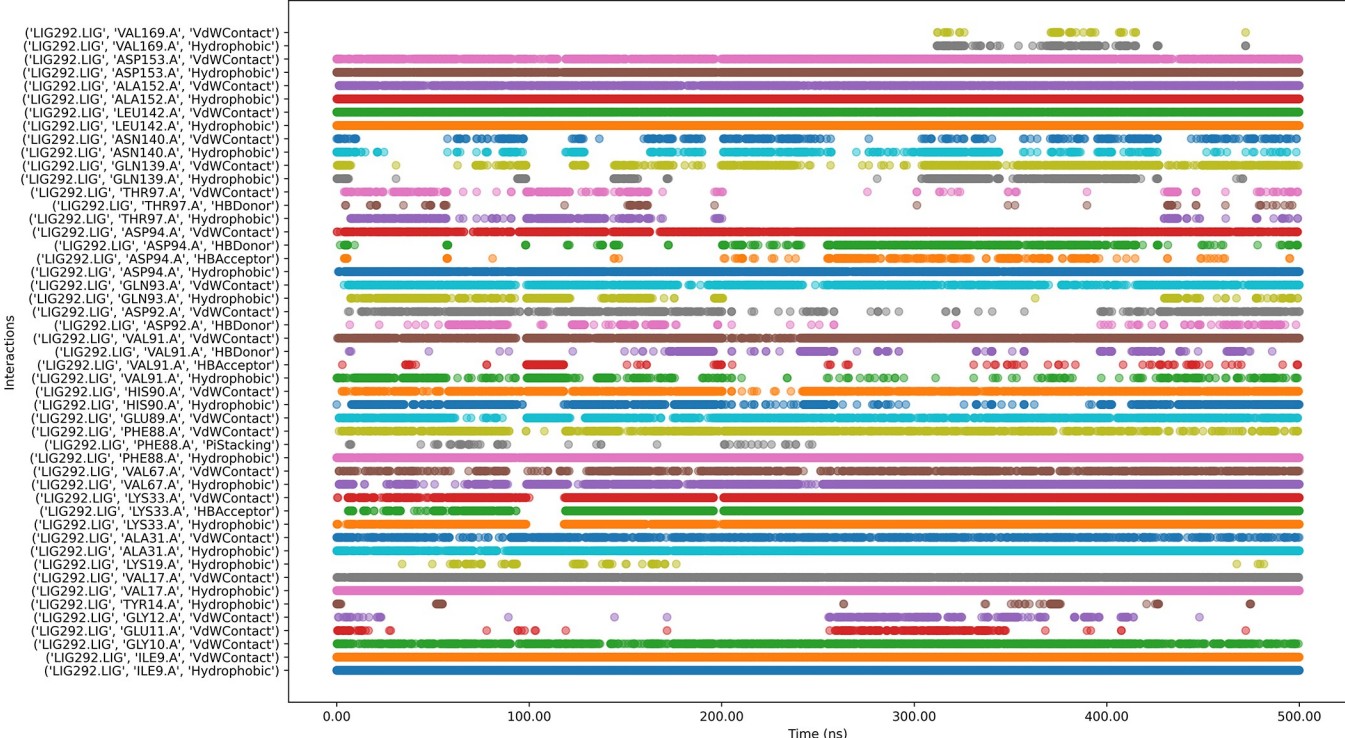

**Fig 12. Interaction timeline of Protein-205813-99-6 complex over time.**

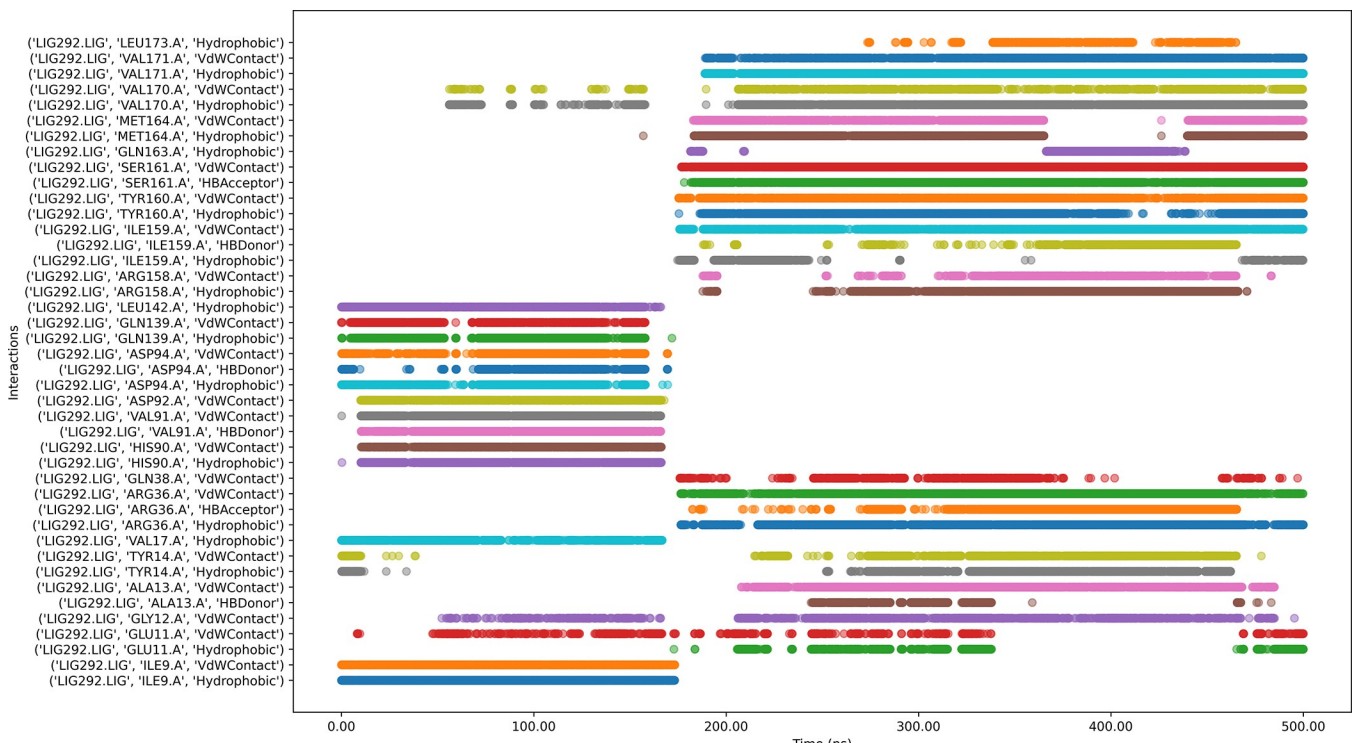

**Fig 13. Interaction timeline of Protein-223130-61-8 complex over time.**

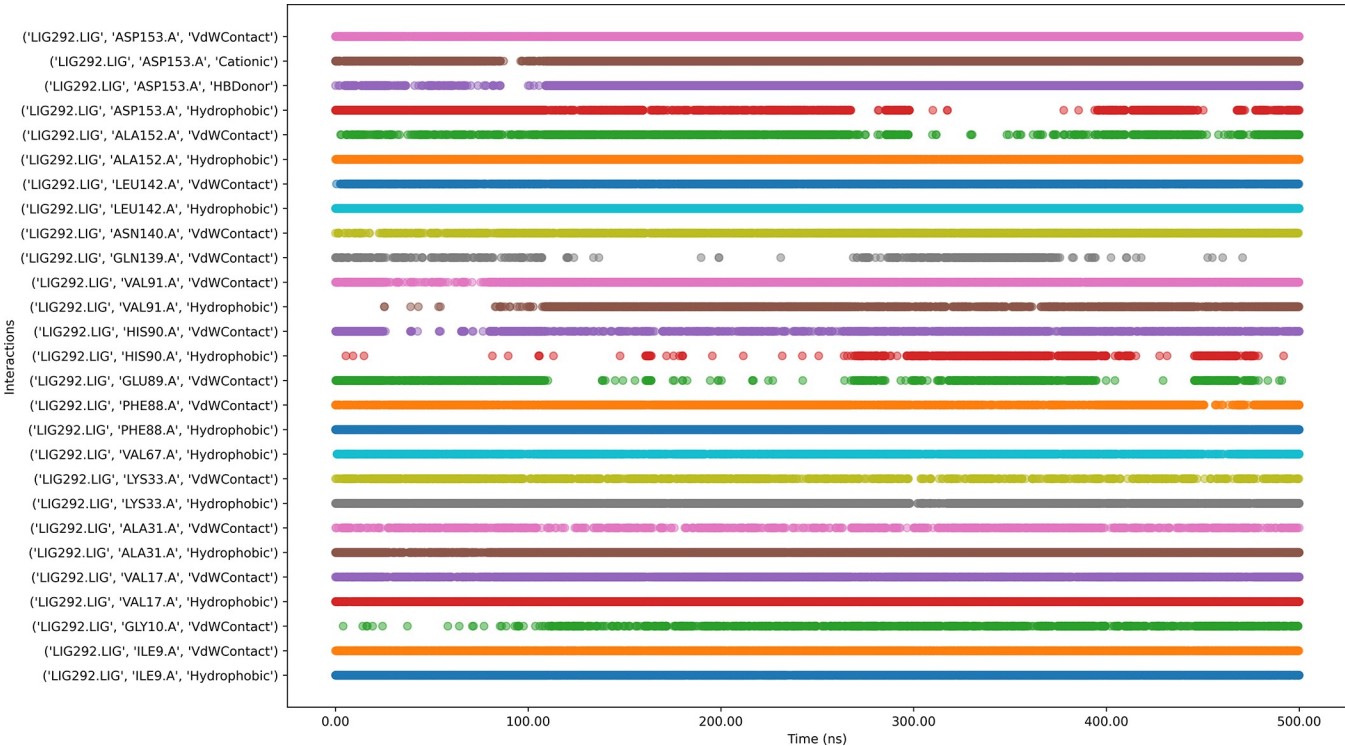

**Fig 14. Interaction timeline of Protein-CMNPD11585 complex over time.**

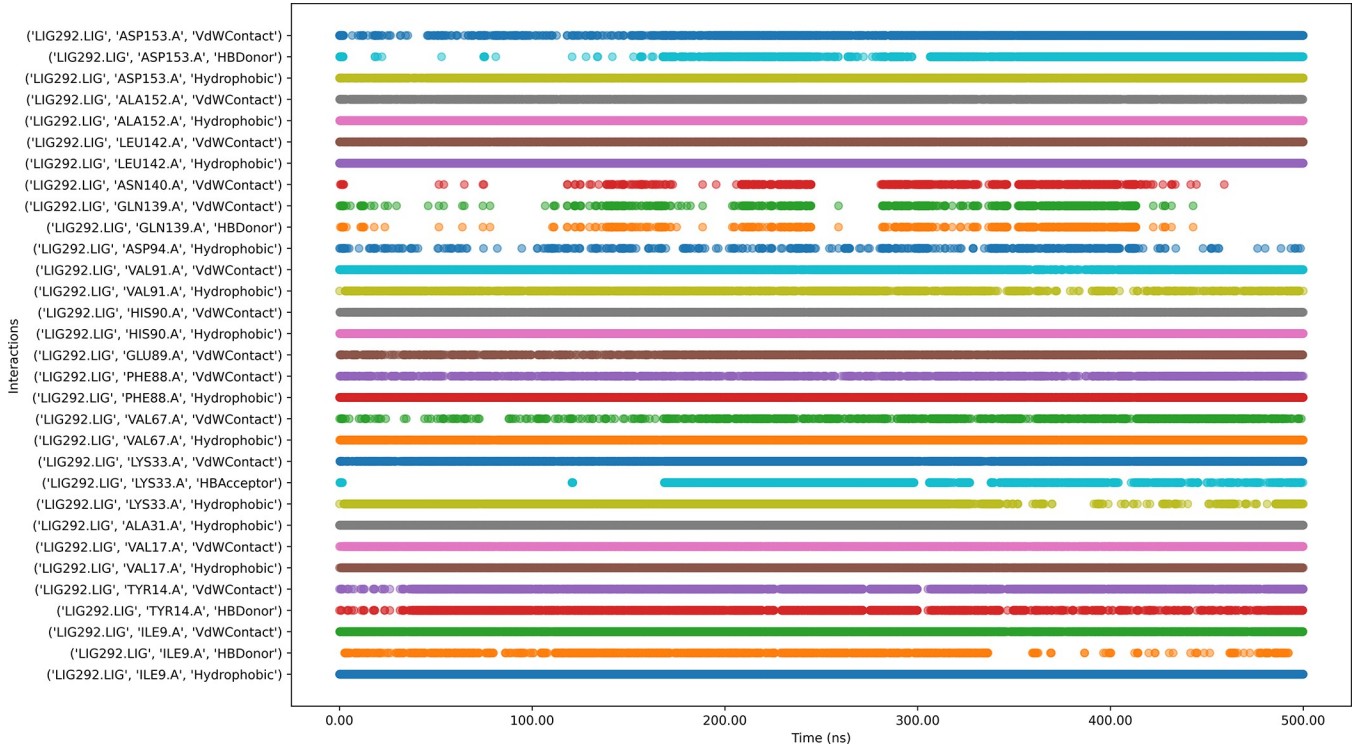

**Fig 15. Interaction timeline of Protein-CMNPD14217 complex over time.**

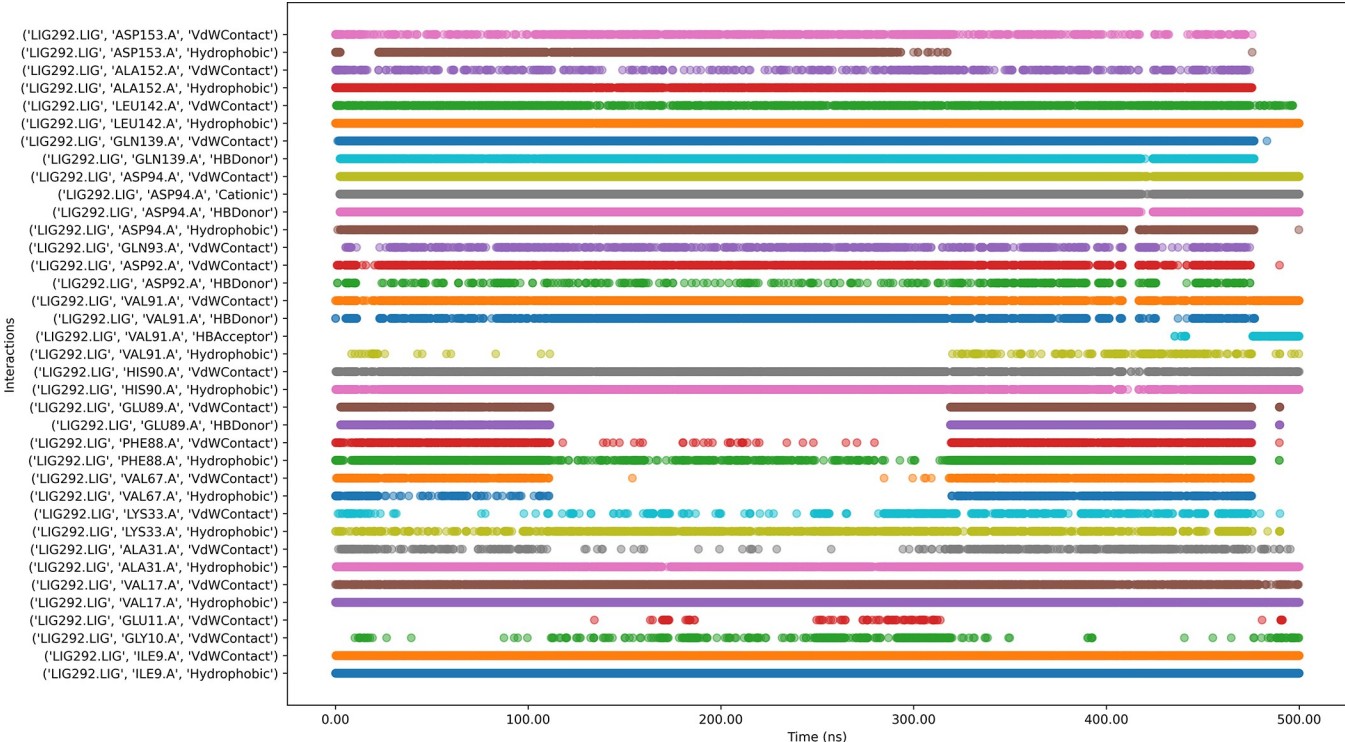

**Fig 16. Interaction timeline of Protein-CMNPD2744 complex over time.**

binding points. The ligand engages in diverse interaction types, including van der Waals contacts, hydrogen bonds, hydrophobic interactions, and water-mediated contacts, suggesting a multifaceted binding mechanism. Notably, ASN140 and GLN139 show more intermittent interactions, potentially indicating areas of flexibility in the binding mode. Some residues, like TYR14, display both van der Waals contacts and hydrogen bonding, highlighting their dual role in ligand stabilization. The persistence of most interactions across the simulation time suggests an overall stable binding pose, though the variability in some interactions (e.g., the hydrogen bonds with GLN139 and TYR14) might indicate areas where ligand optimization could enhance binding affinity or specificity.

Interaction dynamics between CMNPD2744 and protein (Fig 16) indicate persistent interactions throughout the simulation, particularly for residues ASP153, ALA152, LEU142, GLN139, ASP94, HIS90, ALA31, VAL17, and ILE9 which show continuous contact across the entire timeframe. Other residues, such as VAL91, GLN89, PHE88, Val67, and GLY10 demonstrate intermittent but significant interactions. Hydrophobic interactions and van der Waals contacts appear to be predominant, with some hydrogen bonding observed. The pattern suggests a stable binding mode for CMNPD2744 within the binding site, with key residues maintaining consistent interactions throughout the simulation.

Ligand CMNPD7986 with the protein complex (Fig 17) exhibits consistent interactions with VAL169, LEU142, GLN139, Val91, HIS90, ALA31, LYS19, VAL17, and ILE9 throughout the simulation. The ligand engages in diverse interaction types, including van der Waals contacts, hydrogen bonds, hydrophobic interactions, and water-mediated contacts, suggesting a multifaceted binding mechanism. Notably, THR172, VAL170, THR167, ASP153, VAL91, HIS90, TYR14, and GLU11 show more inconstant interactions, potentially indicating areas of flexibility in the binding mode. Some residues display both van der Waals contacts and

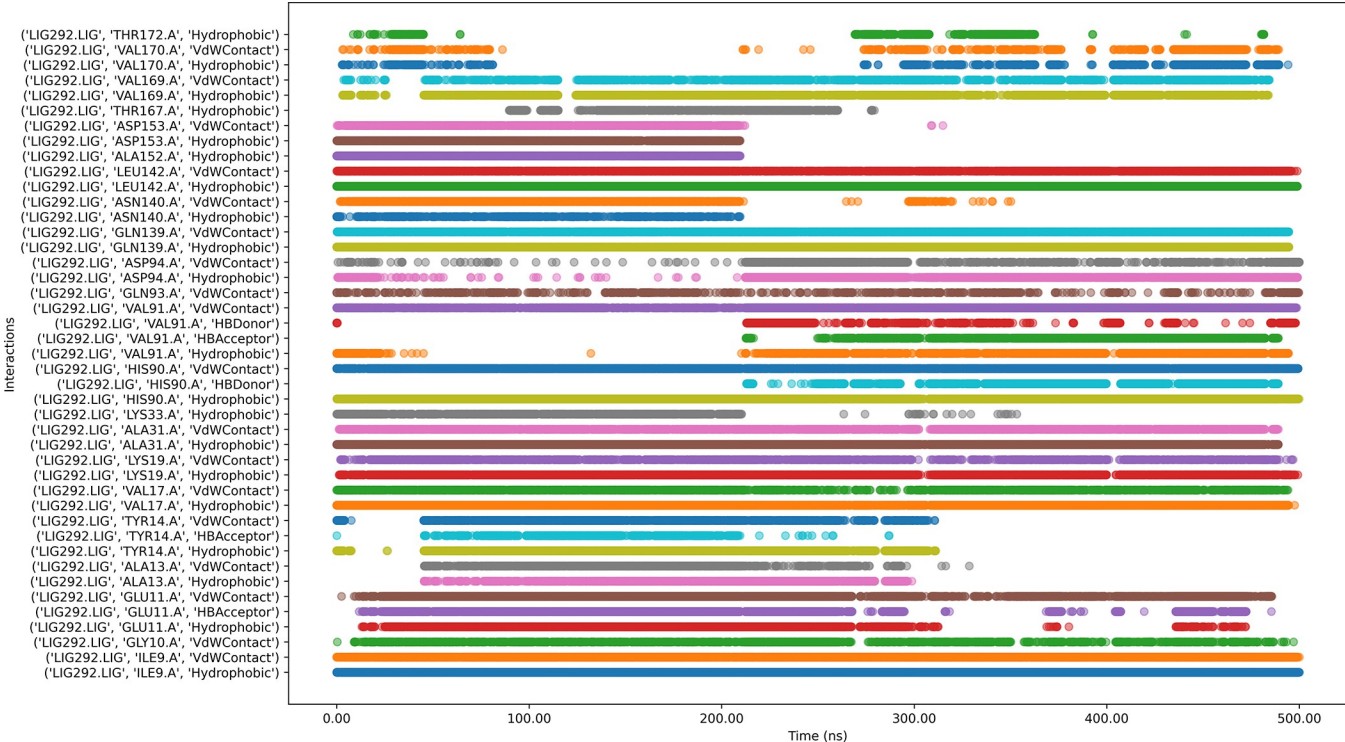

**Fig 17. Interaction timeline of Protein-CMNPD7986 complex over time.**

hydrogen bonding, highlighting their dual role in ligand stabilization. The consistent hydrophobic interactions suggest that the ligand significantly contributes interaction to the hydrophobic pocket of the protein.

**3.6.9 Protein-ligands contact percentage.** A comparative analysis of the protein-ligand contact percentages (Fig 18) reveals that CMNPD2744 and CMNPD14217 exhibit remarkably similar interaction profiles, with both showing strong interactions at key residues: ASP153 (cationic interactions at 98.6% and 93.8% respectively), LEU142 (hydrophobic interactions at 100% and 99.7%), ASP94 (cationic interactions at 75.1% and 89.2%), and VAL91 (hydrophobic interactions at 93.8% and 95.9%). In contrast, CMNPD7986 displays a notably different interaction pattern, characterized by substantially reduced interactions with ASP153 (28.9%) and ASP94 (31.9%), but enhanced interactions with GLU167 (73.7% hydrophobic interaction compared to <25% for the others) and THR171 (39.8% cationic interaction compared to <10% for the others). These pronounced differences in interaction percentages strongly suggest that CMNPD7986 adopts a distinct binding orientation or mode within the protein's binding pocket, potentially due to structural differences or by inducing a different conformational state in the protein. This quantitative comparison of interaction profiles provides crucial insights for structure-based drug design, highlighting specific residues that could be targeted to enhance selectivity or potency in future iterations of these compounds.

**3.6.10 Interaction dynamics with Sankey diagram analysis.** The Sankey diagram (Fig 19) illustrates the interactions between various ligands and amino acid residues of a protein and reveals that ligands such as CMNPD11585, 20581-99-6, CMNPD14217, CMNPD2744, CMNPD7986, and 223130-61-8 interact with certain residues such as THR97: A, ASP153:A, PHE88:A, LYS33:A, VAL67:A, and ALA152:A are common interaction sites for these ligands. The diversity of interactions highlights the protein's flexible binding pockets,

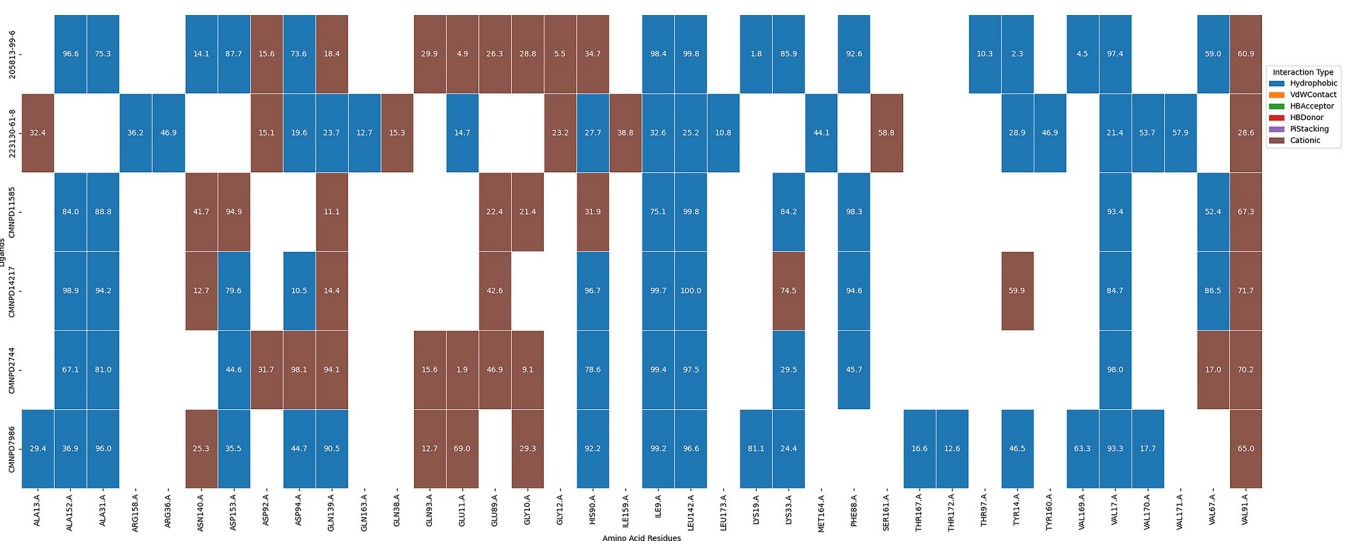

**Fig 18. Delineation of residue-specific protein-ligand interaction profiles.**

accommodating various ligands. Some ligands, such as CMNPD11585 and 20581-99-6, exhibit broad interaction patterns, while others like CMNPD7986 and 223130-61-8 show more selective interactions, suggesting higher affinity for specific residues.

Ligands such as CMNPD2744, 223130-61-8, CMNPD7986, and CMNPD11585 emerge as particularly promising candidates for in-vitro studies due to their demonstrated stable binding and favorable interaction profiles. CMNPD2744 consistently exhibits the lowest RMSD and RMSF values, indicating exceptional stability and minimal conformational fluctuations, which are critical for maintaining strong and specific protein-ligand interactions. Similarly, 223130-61-8 shows a tightly packed conformation with the lowest average Radius of Gyration (Rg) and a well-defined free energy minimum, suggesting a robust and specific binding mode that enhances its potential as a therapeutic agent. These ligands also maintain persistent hydrogen bonds and key residue interactions, further underscoring their stability and specificity. Collectively, the stable binding characteristics and favorable interaction profiles of CMNPD2744, 223130-61-8, CMNPD7986, and CMNPD11585 make them strong candidates for further optimization and development in the pursuit of effective therapeutic agents.

## 3.7 MTT assay

The MTT assay results reveal varying cytotoxic effects of CMNPD2744, 223130-61-8, CMNPD7986, CMNPD11585, and Ribociclib on MCF-7 breast cancer cells. All compounds exhibited concentration-dependent decreases in cell viability, with CMNPD11585 demonstrating the highest potency (IC50 = 0.03 ± 0.002 μM), followed by Ribociclib (IC50 = 0.061 ± 0.006 μM), CMNPD2744 (IC50 = 0.073 ± 0.003 μM), and 223130-61-8 (IC50 = 0.198 ± 0.018 μM).

Notably, CMNPD11585 showed higher and CMNPD2744 showed comparable potency to the standard drug Ribociclib, suggesting their potential as promising anticancer agents. The steep dose-response curves (Fig 20), particularly for CMNPD11585 and CMNPD2744, suggest a narrow therapeutic window. The findings indicate that CMNPD11585 and CMNPD2744 could be promising lead compounds for further research in breast cancer treatment, requiring further preclinical studies to understand their therapeutic efficacy.

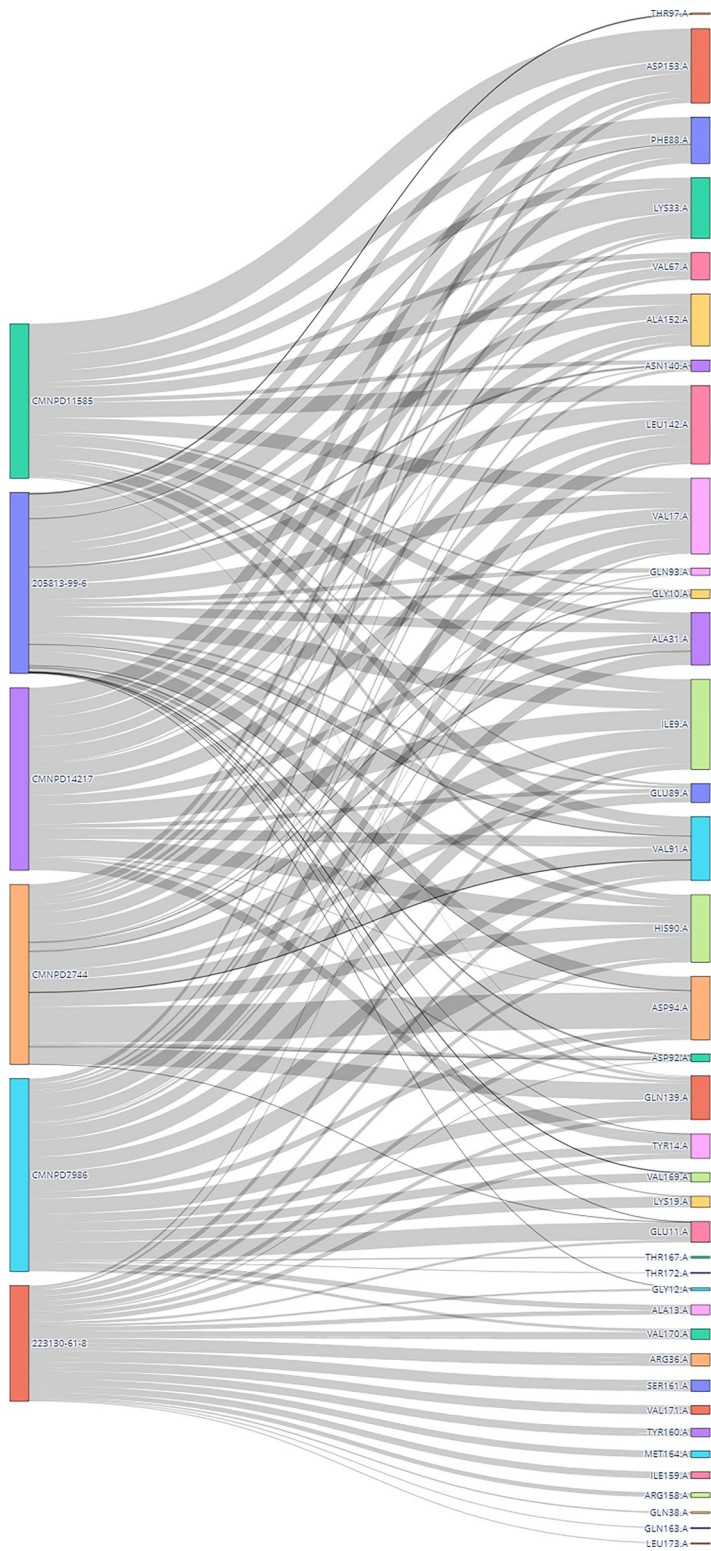

**Fig 19. Sankey diagram visualization of interaction dynamics in protein-ligand complexes.**

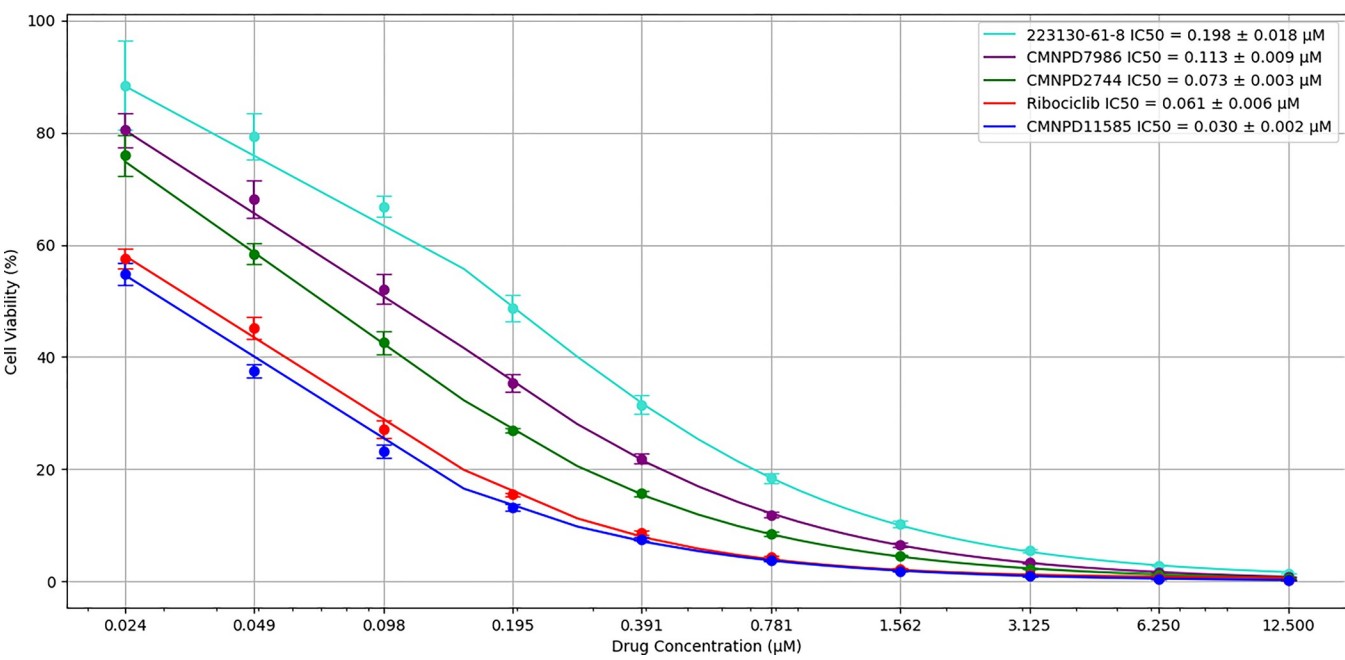

**Fig 20. Dose-response curves of compounds on MCF7 cells.**

## 4. Discussion

Breast cancer remains a critical global health concern, with projections indicating an additional 6.9 million cases worldwide by 2050 [89]. Current therapeutic modalities face significant challenges, including adverse effects, toxicity profiles, and high economic burdens [90]. In response, drug repurposing initiatives, particularly focusing on small compound-based cyclin-dependent kinase (CDK) inhibitors, have gained traction in developing novel therapeutic strategies [91]. The US FDA has approved three CDK4/6 inhibitors for breast cancer treatment: Palbociclib, Ribociclib, and Abemaciclib [92–94]. However, these agents are associated with notable side effects, including neutropenia [95], gastrointestinal disturbances, fatigue [96], and an elevated risk of hepatic dysfunction [97]. Recent years have witnessed a surge in research aimed at identifying novel CDK4/6 inhibitors for breast cancer therapy [98]. Bhattacharya *et al*. 2023 presented their research on flavone compounds as CDK4/6 inhibitors [99]. Baig *et al*. conducted a study on the pharmacological repurposing of Selonsertib as a CDK4/6 inhibitor in 2022. Chukwuemeka *et al*. identified 1,3,4-thiadiazole compounds as CDK6 inhibitors in the same year [100]. Baig *et al*. pioneered the development of innovative pyrazolo [4,3-e] compounds, and Pyrido[1,2-a]Pyrimidine-based cytotoxic drugs targeting CDK4/6 [101]. Nusantoro & Fadlan conducted computational investigations on transition metal compounds of isatinyl-2-aminobenzohydrazone in 2021 [102]. Susanti *et al*. identified several drugs using pharmacophore-based virtual screening, and Gurung *et al*. found bioactive chemicals from Ficus carica L. [103], and Pinanti *et al*. identified biflavonoid chemicals derived from Selaginella doederleinii Hieron [104]. Yousuf and colleagues suppressed CDK4/6 activity in 2020 with Quercetin [105], and Ellagic acid has the potential to be a potent inhibitor promoted by Yousuf's lab [106]. Ullah *et al*. discovered phytochemicals in Clitoria ternatea can effectively inhibit CDK4/6 [107]. Sharma and Kumar demonstrated the effectiveness of Natural Pyridoacridines as CDK4/6 inhibitors in 2016 [108]. This diverse array of studies underscores the ongoing efforts to identify and develop novel CDK4/6 inhibitors from various sources,

including natural products, synthetic compounds, and repurposed drugs. These investigations aim to address the limitations of current therapies and potentially offer more effective and less toxic treatment options for breast cancer patients.

Many of these drug discovery studies did not consider comprehensive analyses of drug-likeness criteria, ADMET properties, toxicity profiles, consensus molecular docking, and molecular dynamics simulations. These assessments are critical in pharmaceutical research, as suboptimal biotransformation and ADME characteristics account for over 40% of preclinical failures [109]. Moreover, inadequate ADMET properties contribute to approximately 60% of new chemical entity attrition in clinical trials [110]. Early prediction of these parameters is therefore essential for streamlining the drug development pipeline and mitigating late-stage failures. Thus, we employed a comprehensive *in-silico* screening approach to identify potent CDK4/6 inhibitors from marine natural products. We selected 2,344 compounds from an initial hit of 9497 based on their drug-likeness and PAINS alert. Further filtration yielded 50 compounds with favorable ADME properties, of which 25 exhibited non-toxic profiles. These 25 candidates underwent consensus molecular docking using seven distinct algorithms to elucidate molecular interactions, optimize conformations at the CDK4/6 active site, and screen 6 compounds. We conducted subsequent molecular dynamics simulations to evaluate structural alterations in ligand-protein complexes. The study found that compounds CMNPD11585 and CMNPD2744 were more stable in their interactions with CDK4/6. They formed stable complexes that kept the integrity of the protein fold. These compounds had lower Solvent Accessible Surface Area (SASA) values and better binding free energies compared to other candidates. This meant they were more compatible with and attracted to the CDK4/6 active site. This strict computational workflow includes drug-likeness criteria, ADME/Tox profiling, consensus docking, and molecular dynamics simulations. The CMNPD11585 is characterized by a polycyclic core structure that serves as a scaffold for key functional groups. The ligand's NH groups, carbonyl moiety, and aromatic ring engage in critical interactions with residues like ASP153, HIS90, and others, through hydrogen bonding, π-π stacking, and hydrophobic contacts (Fig 21).

A positively charged region (N+) forms a significant cationic interaction with ASP153, potentially crucial for binding affinity. The ligand's aliphatic portions contribute to hydrophobic interactions with residues such as VAL91, LEU142, and VAL87. The presence of multiple chiral centers and the overall size and shape of the molecule appear well-suited to the binding pocket. However, the simulation also highlights areas of transient or weak interactions, suggesting opportunities for optimization.

Whereas CMNPD2744's structure features two prominent aromatic rings connected by a central heterocyclic moiety, likely imparting rigidity and defining its overall shape. Key interactions include hydrogen bonding involving the NH group, potentially with residues like ASP153 and GLU89. The two oxygen atoms, likely from methoxy or hydroxyl groups, engage in hydrogen bonding with residues such as VAL91, HIS90, and ASP92. The aromatic rings appear to be involved in hydrophobic interactions and possibly π-π stacking with residues like PHE88, ALA31, and LEU142. A notable feature is the presence of a positively charged nitrogen, forming a crucial cationic interaction with ASP94 (Fig 22). The ligand's size and shape appear well-suited to the binding pocket, with multiple points of contact ensuring a snug fit. The timeline in Fig 16 shows persistent interactions with residues like ASP153, ALA152, and LEU142, indicating stable binding points. However, some interactions, such as those with GLN139 and VAL91, show more variability, The study identifies promising CDK4/6 inhibitors from Marine Natural Products for potential use in medicine. Potential limitations of this study include *in-vitro* studies. Future research should focus on structural optimization, investigating synergistic effects with existing breast cancer therapies. A clear path for clinical translation,

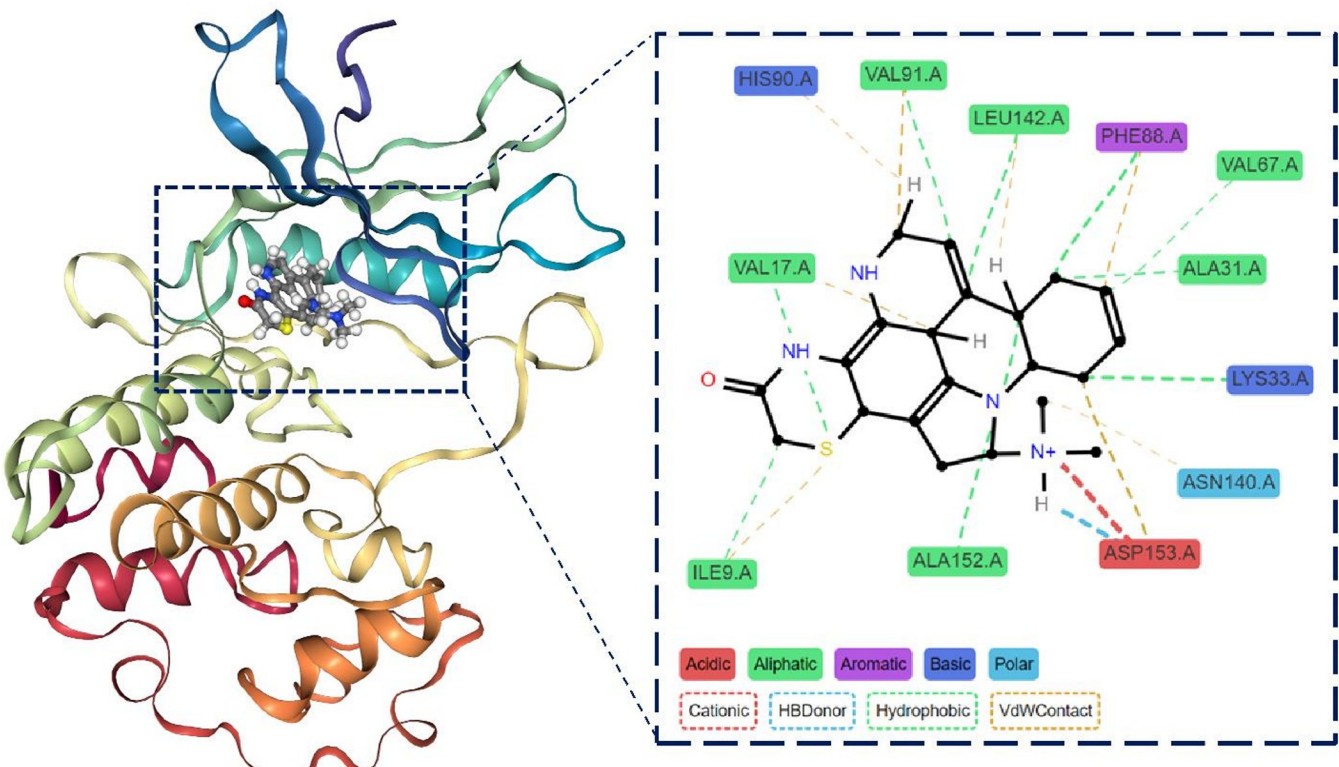

**Fig 21. 3D and 2D interaction between CDK4/6-CMNPD11585.**

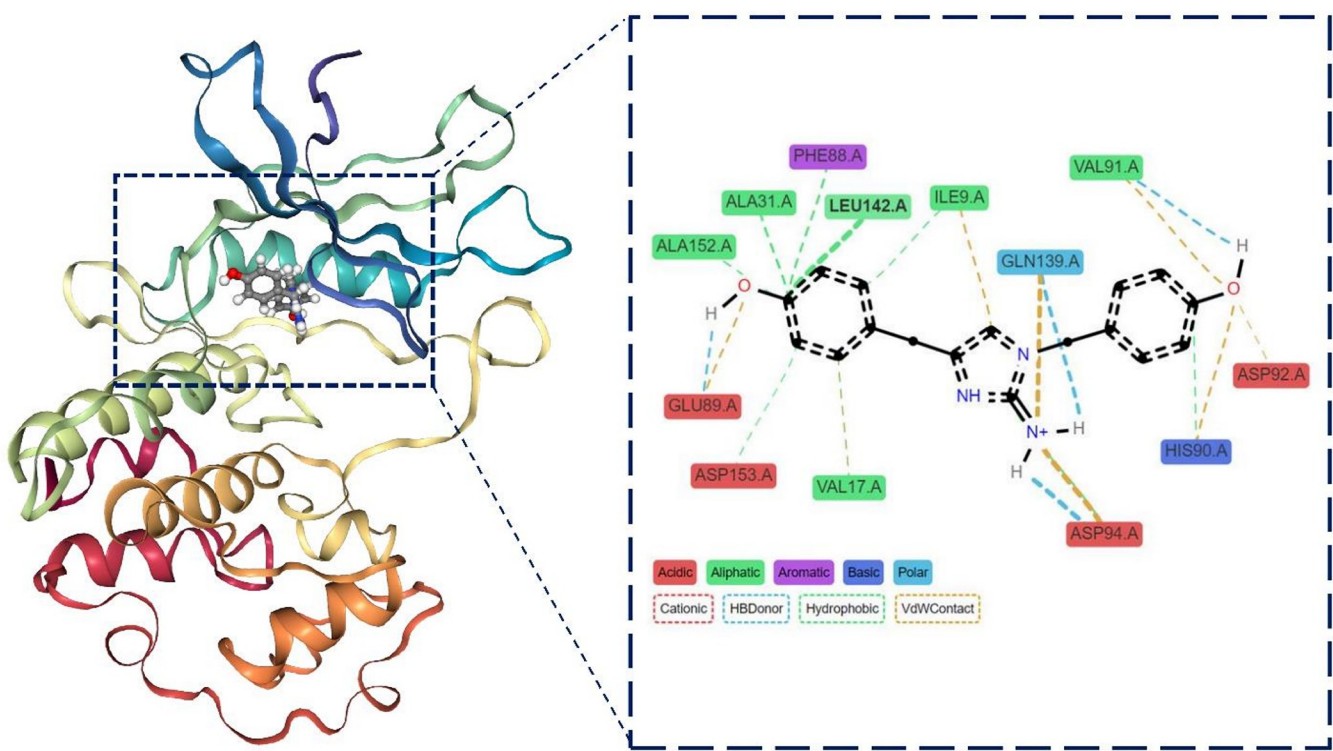

**Fig 22. 3D and 2D interaction between CDK4/6-CMNPD2744.**

including preclinical safety assessments, formulation development, and early-phase clinical trials, is crucial to bridge the gap between *in-silico* and *in-vitro* findings.

## 5. Conclusion

The investigation found two compounds, CMNPD11585 and CMNPD2744, originating from marine natural products. These compounds have the potential to be effective candidates for inhibiting CDK4/6. The *in-silico* studies showed that they bind and stay stable well, and *in-vitro* experiments in the lab confirmed the cytotoxicity of these compounds on MCF7 breast cancer cells very effectively.

## Supporting information

**S1 Graphical abstract.**
(TIF)

## Acknowledgments

We acknowledge Prof. B. Jayaram (Mentor, Supercomputing Facility for Bioinformatics & Computational Biology, IIT Delhi, India) for the human resource training program and inspiration in Drug Discovery.

We also thank the National Supercomputing Mission (NSM) for providing computing resources for 'PARAM Shivay' at the Indian Institute of Technology (BHU), Varanasi. This facility is implemented by C-DAC and supported by the Ministry of Electronics and Information Technology (MeitY) and the Department of Science and Technology (DST), Government of India.

## Declaration

The English in this article has been improved using Large Language Models (LLMs), specifically Llama-2-70B-GGML and Chat Generative Pre-trained Transformer 4.0. We take full responsibility for the content. All LLM-generated text has been thoroughly reviewed, edited, and refined to reflect our original research findings and interpretations accurately.

## Author Contributions

**Conceptualization:** Abhijit Debnath.

**Formal analysis:** Rupa Mazumder.

**Investigation:** Abhijit Debnath, Rajesh Kumar Singh.

**Methodology:** Rajesh Kumar Singh.

**Supervision:** Anil Kumar Singh.

**Writing – original draft:** Abhijit Debnath.

**Writing – review & editing:** Rupa Mazumder, Anil Kumar Singh.

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
