## [Decision Letter · Decision Letter 0]

29 Sep 2024

PONE-D-24-38269Identification of Novel Cyclin-dependent kinase 4/6 Inhibitors from Marine Natural ProductsPLOS ONE

Dear Dr. Debnath,

Thank you for submitting your manuscript to PLOS ONE. After careful consideration, we feel that it has merit but does not fully meet PLOS ONE’s publication criteria as it currently stands. Therefore, we invite you to submit a revised version of the manuscript that addresses the points raised during the review process.

We look forward to receiving your revised manuscript.

Kind regards,

Ahmed A. Al-Karmalawy, PhD

Academic Editor

PLOS ONE

Journal Requirements:

“NO authors have competing interests”

5. Please ensure that you refer to Figure 22 in your text as, if accepted, production will need this reference to link the reader to the figure.

Reviewers' comments:

Reviewer's Responses to Questions

**Comments to the Author**

1. Is the manuscript technically sound, and do the data support the conclusions?

Reviewer #1: Yes

Reviewer #2: Yes

2. Has the statistical analysis been performed appropriately and rigorously? 

Reviewer #1: N/A

Reviewer #2: Yes

3. Have the authors made all data underlying the findings in their manuscript fully available?

Reviewer #1: No

Reviewer #2: Yes

4. Is the manuscript presented in an intelligible fashion and written in standard English?

Reviewer #1: Yes

Reviewer #2: Yes

5. Review Comments to the Author

Reviewer #1: The paper is well presented and the data is well written. My only comment would be to make sure the figures are cleaner. For example, Fig 3 has the labels pf the different rankings overlapped with the outer circle, and Fig 9 and 11 have the box with the legends in transparent, that causes it to overlap with the grid lines on the plot.

Reviewer #2: The study presented provides in silico approach to screen CDK 4/6 inhibitors, while the study and analysis is useful and could be considered for publication, but it needs some corrections.

General Comments:

1) The Introduction section seems non-coherent and should be reorganized. For example, some statements have been repeated and are redundant. Acronyms should be defined whenever they are occuring for the first time in the introduction for readers.

2) Authors should put more emphasis on why marine-based compounds were chosen. Also, was all 9497 identified compounds are marine based?

3) What are color attributes in Table 1 and 2 for consensus molecular docking?

4) Did authors determined protonation states of the CDK4/6 complex??

5) Authors mentioned they simulated 100 ns at physiological conditions. I am rather confused what conditions were used for other 400 ns??

6) Did authors simulated protein in apo state?

7) Kcal/Mol should be changed to kcal/mol (small caps)

8) RMSD plots show the RMSD of the system increases after 400 ns. Would authors comment on this?

9) RMSF figure should be replotted to match the y-axes.

10) How key residues were chosen for RMSF analysis? Why residues 170-180 shows higher RMSF, why they are none of the residues are in key residues? What role these residues play in binding?

11) How the "key residues" were identified.

12) The authors should put cell viability plots?

13) From my understanding authors study aimed to target triple negative breast cancer. Whereas MCF-7 is not triple negative. Would author comment on why MCF-7 cell line was chosen, or should clarify it in the introduction.

14) The text in some plots in too small to be readable. Authors are suggested to increase the font size to improve readability.

6. PLOS authors have the option to publish the peer review history of their article (what does this mean?). If published, this will include your full peer review and any attached files.

Reviewer #1: No

Reviewer #2: No

---

## [Author Response · Author response to Decision Letter 0]

4 Oct 2024

Reviewer # 1:

The paper is well presented and the data is well written. My only comment would be to make sure the figures are cleaner. 

Problem 1: For example, Fig 3 has the labels pf the different rankings overlapped with the outer circle, and Fig 9 and 11 have the box with the legends in transparent, that causes it to overlap with the grid lines on the plot.

Reply 1: We have updated the representation of Fig 3, 9 and 11 as per suggestion. 

Reviewer # 2:

The study presented provides in silico approach to screen CDK 4/6 inhibitors, while the study and analysis is useful and could be considered for publication, but it needs some corrections. General Comments:

Problem 1: The Introduction section seems non-coherent and should be reorganized. For example, some statements have been repeated and are redundant. Acronyms should be defined whenever they are occuring for the first time in the introduction for readers.

Reply 1: We have updated the “Introduction” section as per your suggestion. 

Problem 2: Authors should put more emphasis on why marine-based compounds were chosen. Also, was all 9497 identified compounds are marine based?

Reply 2: Yes, in the updated “Introduction” Section, we have given more emphasis on why marine-based compounds were chosen. Yes The CMNPD database [36], and MNP library [37] both consist of marine-based compounds, thus 9497 identified compounds are marine-based. 

Problem 3: What are color attributes in Table 1 and 2 for consensus molecular docking?

Reply 3: A color gradient from cool (blue) to warm (red) colors is used to visually represent the docking scores, with cooler colors indicating more negative (favorable) binding energies and warmer colors indicating less favorable binding energies. We have updated in the manuscript too.

Problem 4: Did authors determined protonation states of the CDK4/6 complex??

Reply 4: Yes , we did it.

Problem 5: Authors mentioned they simulated 100 ns at physiological conditions. I am rather confused what conditions were used for other 400 ns??.

Reply 5: We have corrected it. 

Problem 6: Did authors simulated protein in apo state?

Reply 6: Yes 

Problem 7: Kcal/Mol should be changed to kcal/mol (small caps)

Reply 7: We have corrected it. 

Problem 8: RMSD plots show the RMSD of the system increases after 400 ns. Would authors comment on this?

Reply 8: Yes, few of them have shown such tendencies probably due to alternative binding mode, we have screened out only those molecules which formed stable complexes. 

Problem 9: RMSF figure should be replotted to match the y-axes.

Reply 9: We have corrected as per suggestion. 

Problem 10: How key residues were chosen for RMSF analysis? Why residues 170-180 shows higher RMSF, why they are none of the residues are in key residues? What role these residues play in binding?

Reply 10: Yes, residues 170-180 shows higher RMSF as because these region is the loop region. Moreover, these regions do not fall in the active site, it is far away from the active site, thus the role these residues play in binding is minimal. 

Problem 11: How the "key residues" were identified.

Reply 11: In the section “2.1.1 Active Site” we have mentioned that binding site residues of the CDK4/6 complex were elucidated through a comprehensive analysis of previously published literature [31–33] and further validated through computational tools CASTp [34] as well as AADS[35]. 

Problem 12: The authors should put cell viability plots?

Reply 12: We have added and updated the manuscript.

Problem 13: From my understanding authors study aimed to target triple negative breast cancer. Whereas MCF-7 is not triple negative. Would author comment on why MCF-7 cell line was chosen, or should clarify it in the introduction.

Reply 13: We have corrected and updated the manuscript accordingly

Problem 14: The text in some plots in too small to be readable. Authors are suggested to increase the font size to improve readability.

Reply 14: We have updated and corrected the plots accordingly.

---

## [Decision Letter · Decision Letter 1]

21 Oct 2024

PONE-D-24-38269R1Identification of Novel Cyclin-dependent kinase 4/6 Inhibitors from Marine Natural ProductsPLOS ONE

Dear Dr. Debnath,

Thank you for submitting your manuscript to PLOS ONE. After careful consideration, we feel that it has merit but does not fully meet PLOS ONE’s publication criteria as it currently stands. Therefore, we invite you to submit a revised version of the manuscript that addresses the points raised during the review process.

We look forward to receiving your revised manuscript.

Kind regards,

Ahmed A. Al-Karmalawy, PhD

Academic Editor

PLOS ONE

Journal Requirements:

Reviewers' comments:

Reviewer's Responses to Questions

**Comments to the Author**

1. If the authors have adequately addressed your comments raised in a previous round of review and you feel that this manuscript is now acceptable for publication, you may indicate that here to bypass the “Comments to the Author” section, enter your conflict of interest statement in the “Confidential to Editor” section, and submit your "Accept" recommendation.

Reviewer #1: All comments have been addressed

Reviewer #2: All comments have been addressed

2. Is the manuscript technically sound, and do the data support the conclusions?

Reviewer #1: Yes

Reviewer #2: Yes

3. Has the statistical analysis been performed appropriately and rigorously? 

Reviewer #1: N/A

Reviewer #2: I Don't Know

4. Have the authors made all data underlying the findings in their manuscript fully available?

Reviewer #1: Yes

Reviewer #2: Yes

5. Is the manuscript presented in an intelligible fashion and written in standard English?

Reviewer #1: Yes

Reviewer #2: Yes

6. Review Comments to the Author

Reviewer #1: (No Response)

Reviewer #2: All of my comments have been addressed. My only suggestion would be uploading high quality plots with readable legends and axis titles. Some figures are fuzzy and labels are too small to be readable.

7. PLOS authors have the option to publish the peer review history of their article (what does this mean?). If published, this will include your full peer review and any attached files.

Reviewer #1: No

Reviewer #2: No

---

## [Author Response · Author response to Decision Letter 1]

31 Oct 2024

To 

The Editor 

PLOS ONE

Subject: Response Letter and Revised Manuscript Submission

Dear Sir,

I thank the reviewers for reviewing our paper submitted to the esteemed journal "PLOS ONE". The reviewers' feedback has been greatly appreciated and we have diligently addressed each comment in our amended article. We have highlighted in yellow color all the text that has been added or changed in the revised manuscript.

Here is our detailed reply, 

Reviewer # 1:

(No Response)

Reviewer # 2:

Problem 1: All of my comments have been addressed. My only suggestion would be uploading high quality plots with readable legends and axis titles. Some figures are fuzzy and labels are too small to be readable.

Reply 1: As per the suggestion of the reviewer, we have uploaed the high quality figures. 

Thanking you

With regards

Abhijit Debnath

Asst. Professor

Noida Institute of Engineering and Technology (Pharmacy Institute), India;

---

## [Editor Report · Decision Letter 2]

1 Nov 2024

Identification of Novel Cyclin-dependent kinase 4/6 Inhibitors from Marine Natural Products

PONE-D-24-38269R2

Dear Dr. Debnath,

We’re pleased to inform you that your manuscript has been judged scientifically suitable for publication and will be formally accepted for publication once it meets all outstanding technical requirements.

Kind regards,

Ahmed A. Al-Karmalawy, PhD

Academic Editor

PLOS ONE
---

## [Editor Report · Acceptance letter]

6 Jan 2025

PONE-D-24-38269R2 

PLOS ONE

Dear Dr. Debnath, 

I'm pleased to inform you that your manuscript has been deemed suitable for publication in PLOS ONE. Congratulations! Your manuscript is now being handed over to our production team.

Kind regards, 

on behalf of

Associate Professor Ahmed A. Al-Karmalawy 

Academic Editor

PLOS ONE